# SpecOffload: Unlocking Latent GPU Capacity for LLM Inference on Resource-Constrained Devices

## Abstract

Efficient LLM inference on resource-constrained devices (*i.e.*, PCs with a single commodity GPU) presents significant challenges in compute and memory utilization. Due to limited GPU memory, existing systems offload model weights to CPU memory, incurring substantial I/O overhead between the CPU and GPU. This leads to two major inefficiencies: (1) GPU cores are underutilized, often remaining idle while waiting for data to be loaded; and (2) GPU memory has a low impact on performance, as reducing its capacity has minimal effect on overall throughput. In this paper, we propose SpecOffload, a high-throughput inference engine that embeds speculative decoding into offloading. Our key idea is to unlock latent GPU resources for storing and executing a draft model used for speculative decoding, thus accelerating inference at near-zero additional cost. To support this, we carefully orchestrate the interleaved execution of target and draft models in speculative decoding within the offloading pipeline, and propose a planner to manage tensor placement and select optimal parameters. Compared with the best baseline, SpecOffload improves GPU core utilization by 4.49× and boosts inference throughput by 2.36×. [Anonymous repo here.](#)

## 1 Introduction

As Large Language Models (LLMs) evolve, their real-world use extends far beyond chatbots to diverse applications including synthetic data generation (Grattafiori et al., 2024), form processing (Chen et al., 2021b), and data wrangling (Narayan et al., 2022). These tasks are characterised by LLMs conducting offline inference in batches over a large number of tokens. For instance, corporations need to process all archives of financial documentation, whilst individuals want to construct knowledge repositories from accumulated materials. In such workloads, higher inference throughput (the number of tokens generated divided by total generation time, token/s) translates into lower total completion time, hence it is the key metric.

Privacy and cost concerns drive these tasks toward LLM deployment on edge servers or PCs, often limited to a single GPU, where GPU memory becomes a major constraint. For example, Mixtral 8x22B (Mix, 2024), with 141 billion parameters, requires at least four state-of-the-art GPUs (H100, 80GB memory) for inference. Offloading is one of the mainstream solutions to memory-constrained inference, transferring most model parameters to more economical, capacious CPU memory and reloading them to GPU memory only when computation demands. There are also methods to overcome the memory bottleneck by compressing the model and KV cache, such as quantization, pruning, sparsification (Krishnamoorthi, 2018; Frankle & Carbin, 2018; Han et al., 2015), *etc.*, which are orthogonal and can be applied on top of offloading.

Our focus is on designing efficient offloading strategies for high-throughput inference on resource-constrained devices with a single GPU. We find that existing offloading approaches do not utilize GPU resources effectively. During offloading, generating each token requires reloading most model parameters from CPU memory to GPU memory for execution (Aminabadi et al., 2022; Eliseev & Mazur, 2023; Fang et al., 2025). Yet I/O speeds substantially lag behind GPU computational capabilities. For instance, under typical NVIDIA RTX 4090, PCIe 4.0×16 conditions, loading a single FFN layer of the Mixtral 8×22B decoder from CPU to GPU consumes 240ms, while the actual

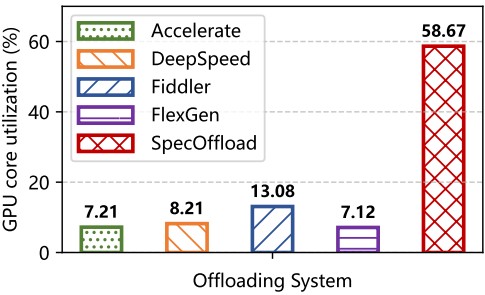 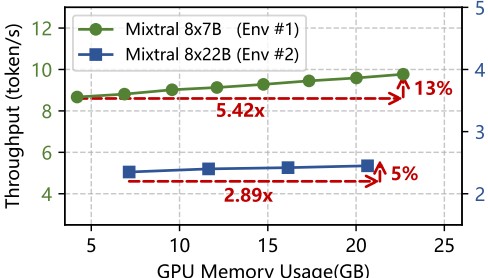

Figure 1: GPU core utilization of SOTA methods during decoding phase. Settings: Mixtral 8x7B, Env #1, SummEval dataset, details in § 5.1.

Figure 2: Impact of GPU memory on throughput during decoding phase. Settings: SummEval dataset, details in § 5.1.

computation on GPU requires merely 0.1ms. Consequently, total inference time is mainly determined by parameter loading time, leaving GPU resources severely underutilized.

To highlight the inefficiencies of existing approaches, we perform a detailed analysis of GPU core and memory utilization. We find that:

- **Underutilization of GPU cores**. As shown in Figure 1, during the decoding phase, the average GPU core utilization of existing methods is only 13% at most. This inefficiency stems from GPU cores frequently remaining long time idle while awaiting parameter loading. To alleviate this issue, existing methods increase the batch size to amortize I/O overhead by loading model parameters once and reusing them across the enlarged batch, thereby improving throughput (Sheng et al., 2023; Fang et al., 2025). However, due to the limitation of GPU memory capacity or CPU computational capabilities, the scalability of the batch size remains inherently limited. The maximum batch size achieved by the SOTA scheme in Figure 1 is only 64, insufficient to bridge the huge gap between I/O and GPU latency (even with this batch size, the gap remains over 10×).

- **Marginal utility of GPU memory**. When model size far exceeds GPU memory, reducing memory usage during decoding phase leads to only marginal throughput degradation. We verify this with FlexGen (Sheng et al., 2023), a leading SOTA offloading framework that computes attention on CPU while keeping FFN layers on GPU. As shown in Figure 2, reducing memory usage by 5.42× for Mixtral-8×7B leads to merely a 13% throughput drop; similarly, a 2.89× reduction for Mixtral-8×22B results in only a 5% decline. The reason is that most parameters cannot permanently reside in GPU memory—each token generation still requires repeatedly loading almost all FFN layers. For example, the leftmost blue point in Figure 2 loads total 56 FFN layers per token, compared to 53 layers for the rightmost point. Consequently, the significant mismatch between model size and GPU memory makes buffering and prefetching ineffective, leading to negligible improvements in overall inference time and throughput even with large memory savings.

To harness GPU compute and memory resources more efficiently, we design SpecOffload, a novel offloading framework that unlocks latent GPU capacity by leveraging speculative decoding (SD). SD is a technique that accelerates generation by employing an auxiliary lightweight draft model to produce multiple candidate tokens, which are subsequently verified in parallel by the target model, enabling the generation of multiple tokens per forward pass (Stern et al., 2018; Leviathan et al., 2023; Chen et al., 2023).

SpecOffload embeds SD into the offloading workflow with nearly zero overhead. The key idea lies in the following two aspects:

- Computing draft model during GPU core idleness: SD requires the draft model to generate multiple candidate tokens in advance. Given the substantial idle periods prevalent in existing frameworks, these intervals can be utilized for completing the draft model's computational tasks.

- Storing draft model uses "low-yield" GPU memory: SD requires loading a draft model into memory for draft generation. We can repurpose "low-yield" memory allocations from existing frameworks to store draft model parameters and its caches instead. For instance, as shown in Figure 2, extracting 17GB of "low-yield" memory allocation provides sufficient capacity for

a draft model such as Mistral 7B (Jiang et al., 2023) to operate normally within the GPU at a small batch.

To support this, SpecOffload designs a comprehensive framework to better utilize both the computational and memory resources of the GPU. SpecOffload determines tensor distribution between GPU and CPU memory through offline Adaptive Tensor Placement (§ 4.2), dynamically schedules computational tasks via the online ParaSpec Planner (§ 4.3), and implements parallel pipelined execution of I/O and computation using the Interleaved Batch Pipeline (§ 4.1).

Our contributions are as follows:

- We conduct a quantitative analysis of GPU resource utilization in representative scenarios and identify key limitations in SOTA frameworks—underutilization of GPU cores and marginal utility of GPU memory, thus reveal a novel perspective for enhancing offloading performance.
- By designing a sub-layer model decomposition and fine-grained scheduling of compute and memory resources, we delicately embedded SD into offloading with virtually zero overhead, thereby increasing GPU core utilization by 4.49 times.
- We evaluate SpecOffload on dense (LLaMA-3.3-70B) and sparse models (Mixtral-8×7B with 46.7B parameters and Mixtral-8×22B with 141B parameters) across four datasets and two hardware settings against five SOTA systems, achieving an average 2.36× throughput improvement. Additional experiments on other hardware further confirm its generality.

## 2 BACKGROUND AND RELATED WORK

### 2.1 SPECULATIVE DECODING

Speculative decoding (SD) is a method for accelerating LLM inference. It adheres to a "Draft-then-Verify" framework: at each decoding step, a lightweight draft model initially proposes multiple candidate tokens (*e.g.*, $(\hat{w}_1, \hat{w}_2, \hat{w}_3, \hat{w}_4)$), which are collectively verified by a larger target model in a single forward pass. Only the valid subset $(w_1, w_2)$ is accepted, after which the target model resumes decoding by independently generating the subsequent token $w_3$ (Stern et al., 2018; Leviathan et al., 2023; Chen et al., 2023). This approach enables the target model to generate multiple tokens per inference step. To further enhance the efficiency of SD, prior research has predominantly explored two avenues: the design of more effective draft models (Zhou et al., 2024; Zhang et al., 2024; Liu et al., 2024) and draft structures (Cai et al., 2024; Miao et al., 2024; Svirschevski et al., 2024).

However, traditional single-batch SD is not well-suited for integration with offloading, as the computations of the draft and target models must be executed sequentially. As a result, the GPU resources remain underutilized during the target model's verification phase. By introducing a dual-batch rotation strategy, SpecOffload enables the verifying and drafting to run concurrently, allowing SD to be seamlessly embedded into the offloading pipeline, while better utilizing GPU resources.

### 2.2 OFFLOADING IN LLM INFERENCE

Offloading is one of the predominant solutions for enabling LLM inference under GPU memory constraints. It entails relocating certain model parameters from the expensive, limited GPU memory to the more cost-effective and abundant CPU memory (Aminabadi et al., 2022; Kamahori et al., 2025; Song et al., 2024; Xue et al., 2025; He & Zhai, 2024; Eliseev & Mazur, 2023). I/O constraints create the primary bottleneck in offloading, as data transfer latency between CPU memory and GPU memory far exceeds the GPU's computation time. Consequently, GPU resources remain underutilized.

In throughput-oriented scenarios, existing methods typically increase the batch size to amortize I/O costs. A prevalent strategy involves altering the model's execution pattern from row-wise to column-wise, allowing each layer's parameters to be loaded once and reused across multiple batches, thereby reducing the per-layer I/O burden (Sheng et al., 2023; Fang et al., 2025). However, this approach is constrained by the available GPU memory and the overhead I/O of KV cache. More recent studies demonstrate that offloading attention computation to the CPU can eliminate KV cache I/O in decoding phase (Cao et al., 2025; Jiang et al., 2024b; Xuanlei et al., 2024; Park & Egger, 2024), but in doing so, CPU compute limitations cap batch scalability. Thus, GPU resources remain significantly underutilized. In this work, by interleaving the draft model's workload into the GPU's idle periods between the target model's layer-wise computations, we improve GPU core utilization by 4.49×.

## 3 SYSTEM OVERVIEW

In this work, we propose SpecOffload. As shown in Figure 3, it employs a two-phase architecture where offline tensor placement and online scheduling collectively determine a unified pipeline across GPU and CPU.

During the offline phase, the target and draft models are deployed across a heterogeneous memory hierarchy. SpecOffload automatically evaluates the hardware ecosystem, measuring CPU and GPU memory capacities, computational performance of CPU and GPU cores, and the bandwidth of data transfer channels between them. These hardware specifications, along with the configurations of models (①), are input to the Adaptive Tensor Placement (§ 4.2) to determine the initial parameter allocation strategy (②).

During the online phase, the hardware configuration and batched inputs (③) are provided to the ParaSpec Planner (§ 4.3), which, based on input length and characteristics, computes a fine-grained pipeline execution plan, including the batch sizes for target model in prefill and decoding phase, batch size for draft model, and the number of candidate tokens to generate (④).

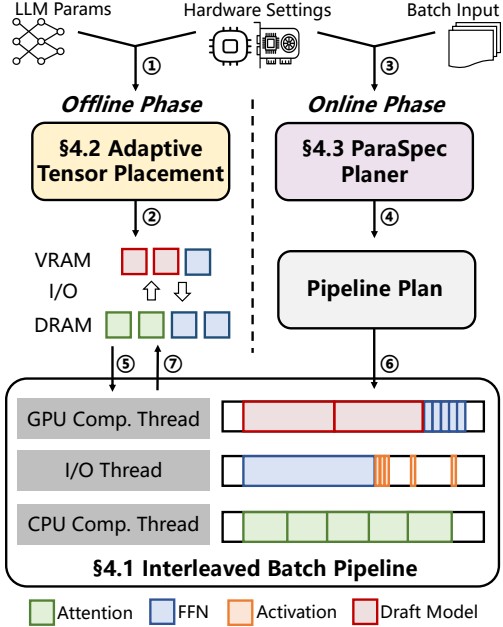

Figure 3: System overview of SpecOffload.

The scheduling results from both offline and online phases (⑤,⑥) collectively determine the Interleaved Batch Pipeline (§ 4.1). The pipeline consists of three main threads: GPU computation, CPU computation, and GPU-CPU I/O. Parameter residency within GPU and CPU memory dynamically adapts in response to the progression of the I/O thread (⑦).

## 4 METHOD

### 4.1 INTERLEAVED BATCH PIPELINE

Motivated by the differing computational demands of the prefill and decoding phases in LLM inference, we introduce the Interleaved Batch Pipeline—a phase-specific pipeline design. During the prefill stage, the target model computation dominates the GPU runtime, and we perform additional memory management before completion. In the decoding stage, we embed speculative decoding into the pipeline by finely interleaving the computations of the two models to fully utilize the GPU cores.

#### 4.1.1 PREFILL PHASE

While our pipeline design for the prefill phase is inspired by the "zig-zag" strategy proposed by FlexGen (Sheng et al., 2023), we extend this approach by tailoring the micro-batch scheduling and parameter management to better support speculative decoding. To minimize the GPU memory footprint of the target model during the offloading stage, at the end of the prefill phase, we offload partial model parameters and the entire KV cache to CPU memory.

#### 4.1.2 DECODING PHASE

During the decoding phase, we build upon the original offloading framework by repurposing the low-yield GPU memory to store the draft model and leveraging GPU idleness to execute it. To enable this, we redesign the entire pipeline at both the model and computation levels. The decoding phase pipeline is illustrated in Figure 4. In summary, our model-level design transforms conventional single-batch speculative decoding into a dual-batch interleaved scheme to facilitate parallel execution. At computation-level, the draft model's workload is finely interleaved into the GPU idle periods between the layer-wise computations of the target model.

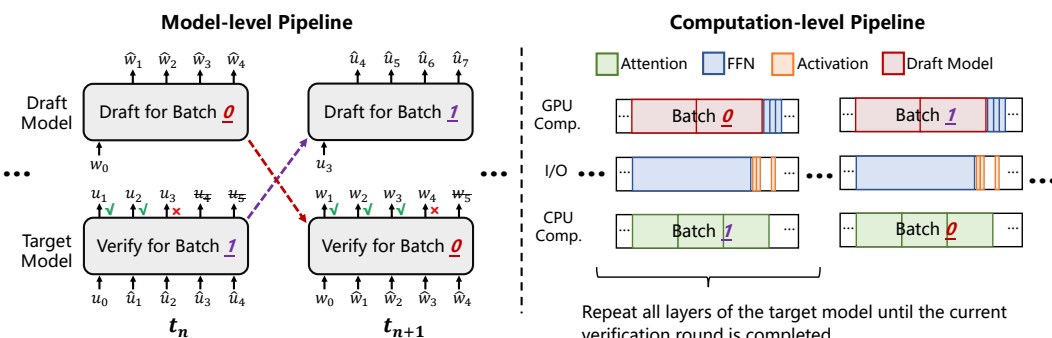

Figure 4: Schematic of the decoding pipeline. At model-level, while the target model validates Batch 1, the draft model concurrently generates tokens for Batch 0 (in time slot $t_n$); the two models then alternate batches (in time slot $t_{n+1}$). At computation-level, the target model's parameters are computed layer by layer. GPU, IO, and CPU are orchestrated to perform distinct, parallelized tasks.

At model-level, the decoding phase involves two batches being processed alternately by the target and draft model. Conventional speculative decoding adopts a single-batch Draft-then-Verify paradigm, as the computations of the draft and target models must be executed sequentially. Consequently, the GPU resources left idle during the target model's verification stage remain unused, while the system suffers from additional overhead caused by switching between models.

To overcome these limitations, we propose a dual-batch interleaved design that enables true model-level parallelism. As shown on the left side of Figure 4, in time slot $t_n$, while the target model verifies Batch 1, the draft model concurrently generates speculative tokens for Batch 0 ($\hat{w}_1, \hat{w}_2, \hat{w}_3, \hat{w}_4$). Once both tasks are completed, the roles switch. In time slot $t_{n+1}$, the target model validates Batch 0 ($w_1, w_2, w_3, w_4$), and the draft model proceeds with Batch 1. This alternating batch rotation continues until the generation is complete. This batch interleaving primarily enables the parallel execution of drafting and verification.

At computation-level, the decoding pipeline involves coordination among three threads: GPU computation, CPU computation, and I/O. The right side of Figure 4 provides an illustrative example. Each layer of the target model is fine-grainedly partitioned into attention and FFN components. For clarity, lightweight components such as normalization layers are omitted from the figure, as their parameter size and computational cost are negligible. As illustrated, for the majority of the inference time, the CPU performs attention computation (current batch), the system transfers same layer's FFN parameters from CPU memory to GPU memory (current batch), and the GPU executes draft model computation (another batch), all in parallel. Upon completion of attention computation on the CPU, the intermediate activations are transferred to the GPU. Finally, once both the parameters and activations are available on the GPU, the remaining computations are quickly completed. After completing all layers of the target model (the end of a round of parallel drafting and verification at model-level), the target and draft models exchange their current batches.

Details such as where model parameters are computed, how I/O is handled, and batch configurations are determined in the § 4.2 and § 4.3.

## 4.2 ADAPTIVE TENSOR PLACEMENT

SpecOffload introduces a novel design for heterogeneous models (target and draft models), by jointly managing the parameter placement across GPU memory, CPU memory, and disk tiers, thus enabling efficient speculative decoding. Adaptive Tensor Placement strategy intelligently assigns tensors to different memory tiers based on real-time resource availability and the current computational task, optimizing memory utilization and mitigating I/O bottlenecks.

We establish tensor prioritization hierarchically by sub-layer, categorizing based on both functional type (attention, KV cache, FFN) and computational phase. Tensors required by the current and next layers of the target model are assigned the highest priority and are preferentially placed in GPU memory. Draft model and its cache are also treated as high-priority and retained in GPU memory during decoding phase. If GPU memory capacity permits, additional parameters are pinned to further

reduce I/O overhead. Remaining tensors are offloaded to CPU memory with moderate priority, leveraging its high bandwidth and low latency, as well as its ability to support certain computations. If CPU memory is exhausted, parameters are further offloaded to disk. When CPU memory is sufficient, $pin\_memory()$ is employed to accelerate GPU-CPU data transfer. A dynamic memory management mechanism is employed to avoid cross-tier memory swaps, ensuring that only CPU memory interfaces with both GPU memory and disk.

The core of the dynamic memory management mechanism is prefetching, which overlaps I/O with computation. For example, while computing attention of layer $i$, GPU memory preloads FFN of the same layer from CPU memory, and concurrently, CPU memory prefetches the parameters of layer $i + 1$ from disk. Dedicated placeholders are reserved in GPU & CPU memory for prefetched tensors.

### 4.3 PARASPEC PLANNER

Interleaved Batch Pipeline section (§ 4.1) outlines our pipeline strategy; however, key parameters—such as the batch sizes of target model during prefill and decoding phase, batch size of draft model, generated draft token number, require careful tuning. To address this, we propose ParaSpec Planner, a parameter specialization module that selects optimal configurations for a given input.

**Planning Goal.** ParaSpec Planner aims to maximize model inference throughput on a given hardware configuration. Throughput is determined by two factors: the total number of tokens generated per batch inference, denoted as $\tilde{N}_{generated}$, and the corresponding generation latency, $T_{generation}$. On consumer-grade hardware, the primary system constraints lie in GPU memory capacity. Therefore, we formulate the problem as a constrained optimization task as follows:

$$\max \ throughput = \max \ \frac{\tilde{N}_{generated}}{T_{generation}}$$

$$s.t. \ gpu \ peak \ memory \ \leq \ gpu \ mem \ capacity \tag{1}$$

**Generated Tokens.** The total number of generated tokens, $\tilde{N}_{generated}$, is the sum of tokens $\tilde{n}_{generated}$ produced over $n_{iter}$ iterations for a batch of size $bs$. However, in our system, speculative decoding introduces randomness, causing the number of tokens generated per input in each iteration to become a random variable. We use the expected value to represent the average number of tokens that pass verification in each iteration.

$$\tilde{N}_{generated} = \sum_{bs} \sum_{n_{iter}} \tilde{n}_{generated} = bs \times n_{iter} \times \mathbb{E}[n_{generated}] \tag{2}$$

**Inference Latency.** The inference latency $T_{generation}$ is determined by the degree of parallelism in the inference pipeline. Due to architectural differences between the prefill and decoding phases in SpecOffload, their latencies must be modeled separately, $T_{generation} = T_{prefill} + T_{decoding}$. Since computation in the prefill phase is primarily GPU-bound, its execution time is independent of batch size and instead depends on the number of computation steps required.

$$T_{prefill} = \left\lceil \frac{bs}{bs_{prefill}} \right\rceil \times T_{target,prefill}^{GPU} \tag{3}$$

During the decoding phase, SpecOffload performs two primary tasks: draft generation for one batch and verification for another. The overall latency is determined by the longer of the two.

$$T_{decoding} = \max(T_{target,decoding}, \ T_{draft}) \tag{4}$$

**Memory Constraints.** GPU memory constraints can likewise be decomposed into those for the prefill and decoding phase. In each phase, the combined memory footprint of model parameters, intermediate activations, and KV cache must not exceed the available GPU memory. In the prefill phase, GPU memory consumption is primarily composed of two parts: the parameter size of the target model, and the KV cache required.

$$V_{prefill} = V_{target,prefill} + V_{target,KVcache} \tag{5}$$

Similarly, in decoding, GPU memory usage consists of the main model parameters, the draft model parameters, and the KV cache used by the draft model.

$$V_{decoding} = V_{target,FFN} + V_{draft} + V_{draft,KVcache} \tag{6}$$

More detailed derivations, please refer to the Appendix A.1. Before using the ParaSpec Planner, a profiling program must be run on the target hardware to collect performance characteristics. However, due to the challenges of hardware measurement, OS-induced variability, and the uncertainty in draft token validity introduced by speculative decoding, such measurements may not fully reflect the actual behavior of SpecOffload during execution. Consequently, while ParaSpec Planner can produce high-quality parameter configurations, further fine-tuning may still be required to achieve optimal performance.

## 5 EVALUATION

Table 1: Hardware Configurations.

|       | Env #1 | Env #2 |
| --- | --- | --- |
| GPU | RTX 4090 | RTX 4090 |
| VRAM | 24G | 24G |
| PCIe | Gen3 x 16 | Gen4 x 16 |
| CPU | i9-10980XE | EPYC 7542 |
| DRAM | 256G | 448G |

Table 2: Dataset Configurations.

|            | HumanEval | C-Eval | SummEval | SAMSum |
| --- | --- | --- | --- | --- |
| $S_{avg}$ | 157.54 | 165.46 | 503.02 | 168.10 |
| $S_{max}$ | 437 | 483 | 783 | 1144 |
| $S_{std}$ | 72.46 | 103.18 | 138.68 | 120.53 |
| Task | Coding | Exam | Summarization | |

### 5.1 EXPERIMENTAL SETUP

**Implementation.** We implement SpecOffload on top of Hugging Face Transformers v4.47.1 (Wolf et al., 2020). We implement pipeline using multiprocessing with shared memory for inter-process vector communication. More details is provided in Appendix A.2.

**Models.** We evaluate SpecOffload on popular and open-source models: Mixtral-8x7B has 47B parameters in total (Jiang et al., 2024a), and Mixtral-8x22B has 141B parameters (Mix, 2024), using Mistral-7B (Jiang et al., 2023) as the draft model. And on LLaMA-3.3-70B-Instruct (lla) with LLaMA-3.1-8B-Instruct (lla) as the draft. All model inference is performed in BF16.

**Hardware.** We evaluate SpecOffload in two different evironments, as shown in Table 1.

**Datasets.** We evaluate SpecOffload on most common LLM benchmarks with varying prompt lengths and tasks. As shown in Table 2: HumanEval (Chen et al., 2021a) with 164 programming problems; C-Eval (Huang et al., 2023), a comprehensive Chinese suite with 13,948 questions; SummEval (Fabbri et al., 2020) with 100 CNN/DailyMail news articles; and SAMSum (Gliwa et al., 2019) with 16k messenger-like conversations and summaries.

**Baselines.** We compare against 4 baseline systems, all designed to address GPU memory limitations.

- Hugging Face Accelerate (Gugger et al., 2022) supports offloading weights of some layers based on the device map. We use its version 1.5.2. Hereinafter referred to as Accelerate.
- DeepSpeed Zero-Inference (Aminabadi et al., 2022) supports offloading the whole weights to the CPU or disk. We use its version 0.16.1. Hereinafter referred to as DeepSpeed.
- FlexGen (Sheng et al., 2023) employs a "zig-zag" inference schedule to increase throughput.
- Fiddler (Kamahori et al., 2025) strategically utilizes both CPU and GPU resources for MoE model inference.

Additionally, FlexGen only supports OPT models, we adapted FlexGen to the experimental models models while adhering to its original offloading strategy[1].

**Metrics.** Throughput (token/s) is calculated as the number of tokens generated divided by total genration time (prefill time + decoding time).

### 5.2 END-TO-END THROUGHPUT

Figure 5 illustrates the end-to-end throughput of five approaches across two environments and four datasets[1]. SpecOffload delivers significant throughput gains over the best baseline, FlexGen, achieving 2.55× on Mixtral-8x7B (Env #1), 2.57× on Mixtral-8x22B (Env #2), and 1.96× on LLaMA-3.3-70B (Env #1). The results show that SpecOffload performs well on both dense and sparse models.

SpecOffload maintains high throughput across diverse hardware and moderate GPU memory increases (see Appendix A.3.4 and Appendix A.3.3).

---

[1]Fiddler is designed for MoE models and does not work with LLaMA.

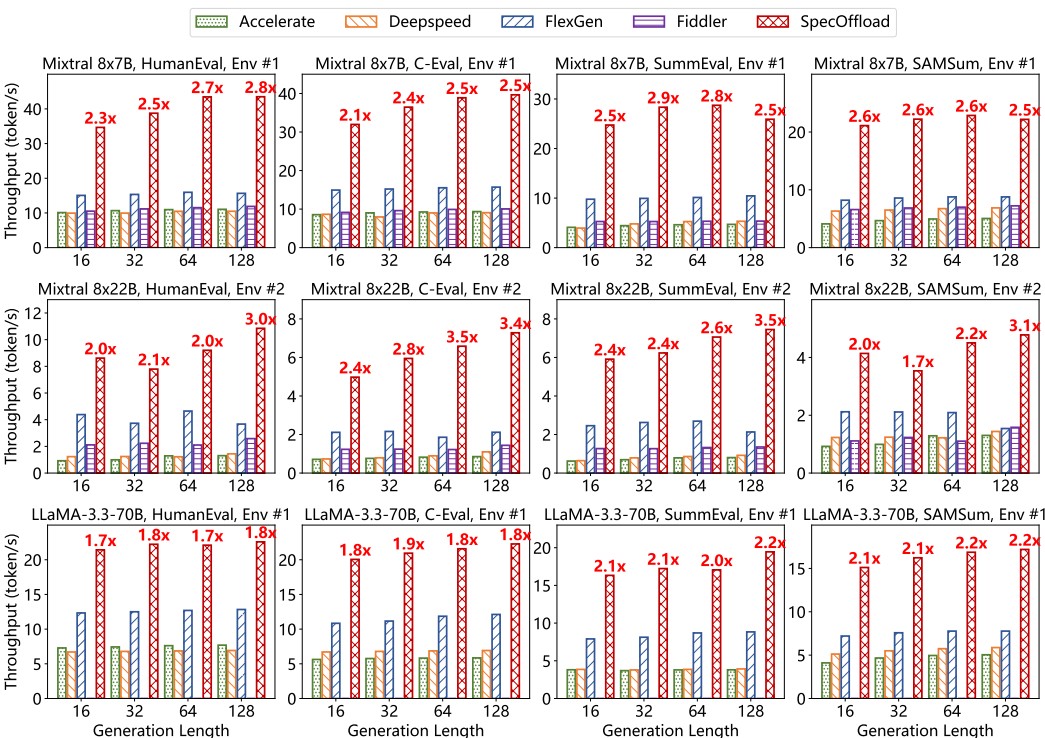

Figure 5: End-to-end comparison between SpecOffload and baselines in different scenarios.

## 5.3 ABLATION STUDY OF THE DRAFT MODEL

To validate the generality of SpecOffload, we replaced the draft models with different families and smaller models. Results in Table 3 show that: among the draft models listed in table, SpecOffload still yield at least 70% improvement. Both small-scale models and models from different families can serve as suitable draft models.

Table 3: Ablation study of the draft model. Target Model Mixtral 8x7B, HumanEval, Env #1, generate 16 token.

| Draft Model | Throughput (token/s) | Improvement (vs. No SD) |
|---|---|---|
| No SD | 16.468 | / |
| LLaMA-3.2-3B | 28.014 | 70% |
| LLaMA-3.1-8B | 30.752 | 87% |
| Mistral 7B | 34.665 | 110% |

## 5.4 ABLATION STUDY OF PROPOSED TECHNIQUES

Table 4: Ablation study of proposed techniques on HumanEval dataset. The gray tuple denotes a policy (Prefill batch size, decoding batch size, draft batch size, draft max new tokens).

| | All optimizations | No policy search | Serial SD | No SD |
|---|---|---|---|---|
| 8x7B | 34.665 (80, 256, 10, 6) | 15.869 (80, 160, 6, 1) | 15.005 (80, 256, 32, 6) | 16.468 (80, 192, x, x) |
| 8x22B | 8.617 (32, 128, 6, 4) | 4.510 (32, 192, 6, 8) | 5.264 (32, 128, 32, 8) | 4.108 (16, 64, x, x) |

We isolate the impact of each technique , as shown in Table 4. On Mixtral-8x7B, SpecOffload uses a target prefill batch of 80, decoding batch of 192 (rotating batches give a total of 384), with a draft batch of 8 generating 8 tokens per iteration. "No policy search" shows the cost of a random strategy. Embedding SD in the pipeline is beneficial, as naive Serial SD with offloading adds I/O overhead. Each design component proved effective. Ablations on other datasets confirm this, in Appendix A.3.6.

Appendix A.3.1 shows the importance of the policy, while Appendix A.3.2 demonstrates that our Paraspec planner can achieve 93% of the optimal policy's performance.

## 5.5 EFFECTIVENESS ANALYSIS

We employ NVIDIA Nsight (nsi, 2025) to monitor GPU core utilization and memory consumption during the decoding phase of Mixtral 8×7B in Env #1 on the SummEval dataset. As depicted in

Table 5: Runtime breakdown (s). "P"/"D": Prefill / Decoding Phase. Compute(G,T/D): GPU time for target / draft model; Compute(C): CPU time for target model. Cache(G→C): KV cache transfer from GPU to CPU.

| | Phase | Total | Compute(G,T) | Compute(G,D) | Compute(C) | Weight(R) | Cache(G→C) |
|---|---|---|---|---|---|---|---|
| 8×7B, | P | 183.28 | 79.62 | 0 | 0 | 123.48 | 39.05 |
| Env #1 | D | 569.21 | 35.34 | 489.02 | 531.23 | 236.2 | 0 |
| 8×22B, | P | 280.42 | 42.22 | 0 | 0 | 166.45 | 91.06 |
| Env #2 | D | 794.26 | 27.34 | 345.93 | 746.38 | 262.64 | 0 |

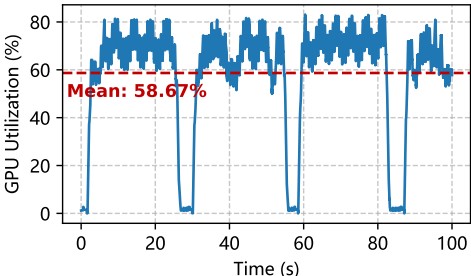

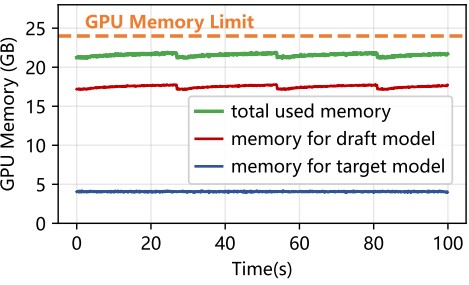

Figure 6: Decoding phase GPU core utilization.     Figure 7: Decoding phase memory consumption.

Figure 6, the average core utilization reaches 58.67%, attains at least 4.49x higher than SOTA. Figure 7 shows a periodic pattern in the draft model's GPU memory: usage rises gradually, drops sharply, then idles for 2 seconds, matching Figure 6 where computation lasts 26 seconds followed by 2 seconds idle. Appendix A.3.5 details our efficient use of GPU memory.

Table 5 breaks down runtime for Mixtral-8x7B (Env #1) and 8x22B (Env #2) on SummEval, profiling GPU, I/O, and CPU with overlapping disabled. Results show our method effectively overlaps compute and I/O, indicating the pipeline works as intended.

## 5.6 LOAD TO DISK

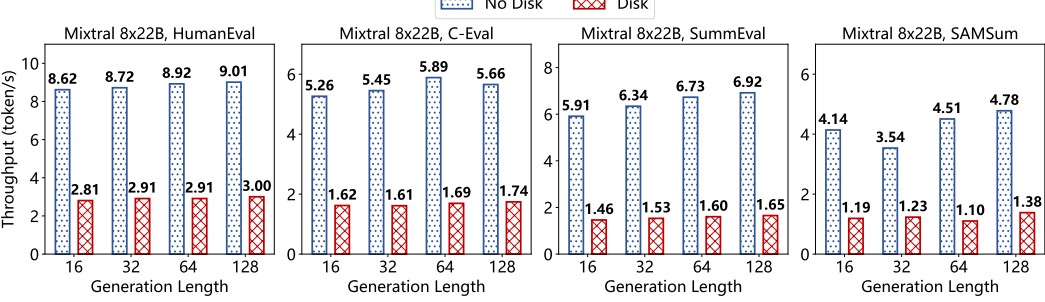

Figure 8: Mixtral-8×22B throughput with vs. without disk offloading (No Disk: Env #2, enough CPU memory; Disk: Env #1, limited CPU memory).

We further conducted experiments in Env #1 by extending the offloading to disk. The disk read and write speeds are 3.5GB/s and 1.7GB/s, respectively. As shown in Figure 8, under these memory-limited conditions, load to disk SpecOffload attains 29.3% of its original throughput, still efficient.

## 6 CONCLUSION

We identify two key inefficiencies in existing offloading frameworks for LLM inference: underutilization of GPU cores and marginal utility of GPU memory. To address these, we propose SpecOffload, which embeds speculative decoding into offloading with virtually zero overhead by leveraging idle GPU time and "low-yield" GPU memory. Experiments show up to 2.36× throughput gains over the best baseline, demonstrating the effectiveness of our approach for high-throughput LLM inference on resource-constrained devices.

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

## A    TECHNICAL APPENDICES AND SUPPLEMENTARY MATERIAL

### A.1    PARASPEC PLANNER

**Planning Goal.**

ParaSpec Planner aims to maximize model inference throughput on a given hardware configuration. Throughput is determined by two factors: the total number of tokens generated per batch inference, denoted as $\tilde{N}_{generated}$, and the corresponding generation latency, $T_{generation}$. On consumer-grade hardware, the primary system constraints lie in GPU memory capacity. Therefore, we formulate the problem as a constrained optimization task as follows:

$$\max \ throughput = \max \frac{\tilde{N}_{generated}}{T_{generation}}$$
$$s.t. \ gpu\,peak\,memory \ \leq \ gpu\,mem\,capacity \tag{7}$$

**Generated Tokens.**

The total number of generated tokens, $\tilde{N}_{generated}$, is the sum of tokens $\tilde{n}_{generated}$ produced over $n_{iter}$ iterations for a batch of size $bs$ . In conventional decoding without speculation, each input generates exactly one token per iteration, making $\tilde{n}_{generated}$ a constant. However, in our system, speculative decoding introduces randomness, causing the number of tokens generated per input in each iteration to become a random variable. As a result, $\tilde{n}_{generated}$ cannot be expressed deterministically and is instead characterized by its expected value. Let $\tilde{n}_{generated}$ denote the actual number of tokens generated for a single input, then $\tilde{n}_{generated} = \mathbb{E}[n_{generated}]$, as shown in Equation 9.

$$\tilde{N}_{generated} = \sum_{bs} \sum_{n_{iter}} \tilde{n}_{generated} \tag{8}$$
$$= bs \times n_{iter} \times \mathbb{E}[n_{generated}] \tag{9}$$

To characterize the distribution of $\tilde{n}_{\text{generated}}$, we model the speculative decoding process. In each iteration, the draft model generates a candidate sequence of $n_{\text{cand}}$ tokens, which is then verified by the target model. The target model returns the longest correct prefix of the candidate sequence and subsequently generates one additional correct token. The number of tokens correctly predicted by the draft model ranges from 0 to $n_{\text{cand}}$, so $\tilde{n}_{\text{generated}}$ follows a distribution over the set $\{1, \ldots, n_{\text{cand}} + 1\}$.

We assume that the probability of the draft model correctly predicting a single token is $p$, and that these predictions are independent across positions. Under this assumption, the probability that the main model accepts exactly $k$ tokens is given by the probability that the first $k - 1$ tokens are correct and the $k$th is incorrect, as shown in Equation 10. If $k = n_{\text{cand}} + 1$, it corresponds to the draft model correctly predicting the entire candidate sequence, this probability distribution is formalized in in Equation 11.

$$\mathbb{P}[n_{generated} = k] = p^{k-1} \cdot (1 - p_{cand}), \quad k = 1, \cdots, n_{cand} \tag{10}$$

$$\mathbb{P}[n_{generated} = k] = p^{k-1}, \quad k = n_{cand} + 1 \tag{11}$$

The expected value $\mathbb{E}[n_{\text{generated}}]$ is derived in Equation 12. Thus, the total number of tokens generated by the model, $\tilde{N}_{\text{generated}}$, is expressed as a function of $bs$, $n_{\text{iter}}$, $n_{\text{cand}}$, and $p$.

$$\mathbb{E}[n_{generated}] = \sum_{k=1}^{n_{cand}+1} k \cdot \mathbb{P}[n_{generated} = k]$$
$$= \frac{1}{1-p}[n_{cand}p^{n_{cand}+2} - (n_{cand} + 1)p^{n_{cand}+1} + 1] \tag{12}$$

**Inference Latency.**

The inference latency $T_{\text{generation}}$ is determined by the degree of parallelism in the inference pipeline. As SpecOffload exhibits distinct behaviors in the Prefill and decoding stages, their latencies must be computed separately, in Equation 13.

$$T_{generation} = T_{prefill} + T_{decoding} \tag{13}$$

In the Prefill stage, loading the full KV cache for all $bs$ inputs would exceed GPU memory capacity. Therefore, SpecOffload partitions the batch into small Prefill batch $bs_{prefill}$ . Since computation in the Prefill stage is primarily GPU-bound, its latency is independent of the Prefill batch size and instead determined by the number of iterations required, as formalized in Equation 14.

$$T_{prefill} = \left\lceil \frac{bs}{bs_{prefill}} \right\rceil \times T^{GPU}_{target,prefill} \tag{14}$$

In each iteration, the processing time per $bs_{\text{Prefill}}$ is primarily determined by parameter I/O ($T^{C2G}_{para}$) and computation ($T^{GPU}_{target,comp}$), with I/O time significantly exceeding computation time in the offloading scenario, as shown in Equation 15.

$$T^{GPU}_{target,prefill} = T^{C2G}_{para} + T^{GPU}_{target,comp} \approx T^{C2G}_{para} \tag{15}$$

In the decoding stage, SpecOffload performs two primary tasks in parallel: draft generation for one batch and verification for another. The overall latency is thus determined by the slower of the two tasks in Equation 16.

$$T_{decoding} = \max(T_{target,decoding}, \ T_{draft}) \tag{16}$$

The draft generation task incurs a latency equal to the time required to execute the draft model inference entirely on the GPU. Similarly, due to memory constraints, the draft model must also divide each batch into smaller sub-batches $bs_{draft}$ for generation, $T^{GPU}_{draft}$ is the time for one-batch generation. Each generation step can be further decomposed into Prefill and decoding stages, as shown in Equation 17.

$$\begin{aligned} T_{draft} &= \left\lceil \frac{bs}{bs_{draft}} \right\rceil \times T^{GPU}_{draft} \\ &= \left\lceil \frac{bs}{bs_{draft}} \right\rceil \times [T^{GPU}_{draft,prefill} + (n_{cand} - 1)T^{GPU}_{draft,decoding}] \end{aligned} \tag{17}$$

For the verification task, based on SpecOffload 's pipeline design, each decoder layer's FFN computation depends on both the output of the Attention module and the loading of FFN parameters. Attention computation and FFN loading are executed in parallel threads, with the completion time determined by their maximum. Subsequently, the GPU performs FFN computation, which is significantly faster than parameter loading. Therefore, its latency can be expressed as in Equation 18.

$$\begin{aligned} T_{target,decoding} &= n_{layer} \times [\max(T^{CPU}_{target,Attention}, T^{C2G}_{target,FFN}) + T^{GPU}_{target,FFN}] \\ &\approx n_{layer} \times [\max(T^{CPU}_{target,Attention}, T^{C2G}_{target,FFN})] \end{aligned} \tag{18}$$

Importantly, since the Attention module is offloaded to the CPU, its runtime becomes dependent on the sub-batch size, and is modeled accordingly in Equation 19.

$$T^{CPU}_{target,Attention} = n_{cand} \times bs \times t^{CPU}_{target,Attention} \tag{19}$$

**Memory constraint.**

The GPU memory capacity constraint can similarly be decomposed into separate constraints for the prefill and decode phases. In both phases, the total GPU memory consumption—including model parameters, intermediate activations, and KV cache—must not exceed the available GPU memory, which directly impacts the feasible batch size per inference.

In the prefill phase, GPU memory usage primarily consists of two components: the memory footprint of the main model parameters ($V_{\text{main}}$) and the KV cache required during prefill ($V_{m,\text{KVcache}}$). Since the KV cache for all $bs$ inputs would exceed GPU memory capacity, the batch is partitioned into sub-batches of size $bs_{\text{prefill}}$. As a result, the KV cache footprint in the prefill phase only accounts for $bs_{\text{prefill}}$ inputs, as formalized in Equation 20.

$$
\begin{aligned}
V_{prefill} &= V_{target,prefill} + V_{target,KVcache} \\
&= V_{target,prefill} + bs_{prefill} \times l_{input} \times v_{target,KVcache}
\end{aligned}
\tag{20}
$$

Similarly, GPU memory usage in the decode phase consists of three components: the main model parameters loaded into GPU memory, the draft model parameters, and the KV cache used by the draft model. According to SpecOffload's offloading strategy, only the MoE FFN parameters from the main model are loaded into GPU memory, whereas the draft model is fully resident in GPU memory. Therefore, the total GPU memory footprint during decoding is characterized in Equation 21. To satisfy the memory constraint, the batch of $bs$ inputs is partitioned into sub-batches of size $bs_{\text{assist}}$, as defined in Equation 22.

$$
V_{decoding} = V_{target,FFN} + V_{draft} + V_{draft,KVcache}
\tag{21}
$$
$$
= V_{target,FFN} + V_{draft} + bs_{draft} \times (l_{input} + n_{generated}) \times V_{draft,KVcache}
\tag{22}
$$

## A.2 IMPLEMENTATION

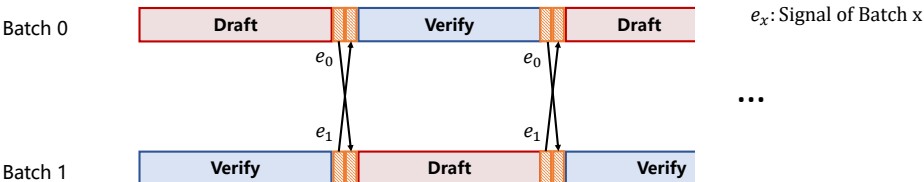

Figure 9: Implementation from the perspective of Interleaved batches.

Our implementation is based on modifications to HuggingFace Transformers Wolf et al. (2020), version 4.47.1.

To implement the SpecOffload pipeline, we adopt a hybrid parallelism strategy that combines process-level and thread-level parallelism. As shown in Figure 9, the input sequence is split into two interleaved batches, which alternate between draft generation and large-model verification. Each batch is processed on a separate thread, with synchronization managed via inter-thread events. After completing its generation and verification task in each iteration, a thread signals its completion and waits for the other thread to do the same. The next iteration begins only after both threads have finished the current one. This design enables parallel execution of Batch 0's draft generation and Batch 1's verification. However, from the perspective of a single batch (Batch 0 or Batch 1), the draft and verify stages remain sequential.

In this design, the draft model performs full-sequence autoregressive inference on GPU during the generation stage, while the large model remains computing on CPU to avoid resource contention. Given that the draft model executes strictly in a sequential manner—processing only one batch at any time—parallel instantiation is unnecessary. To further minimize GPU memory consumption, only a single copy of the draft model is loaded and isolated within a dedicated auxiliary process. Communication between this draft model process and the main process hosting the large model is established via shared memory, enabling low-latency data transfer, while inter-process events are employed to enforce strict execution ordering and synchronization.

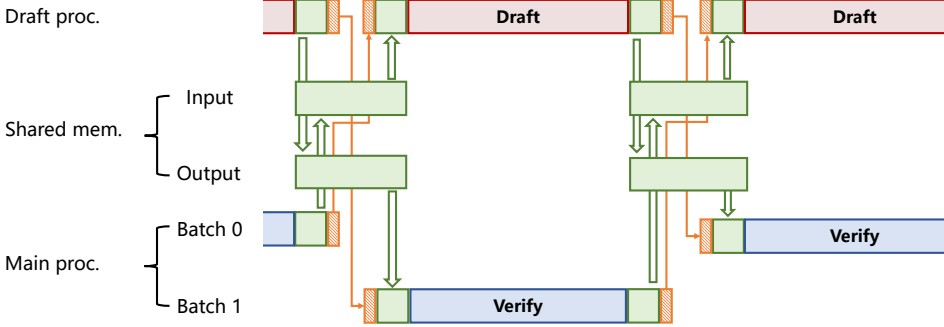

Figure 10: Inter-process communication diagram. Orange blocks represent the send/receive modules within each process, while green blocks indicate the inter-process communication modules. "Input" and "Output" are defined with respect to the Draft process.

As shown in Figure 10, inter-process communication is centered around two shared memory regions, with tokens as the primary data exchanged. The Draft process writes generated draft tokens to the output shared memory for consumption by the Main process, and reads verified tokens from the input shared memory region. When a thread in the Main process reaches the verify stage and requires draft tokens, it waits for the draft model to signal availability, then reads from shared memory. Upon finishing computation, the draft model signals completion and readiness for the next task using event flags, ensuring proper synchronization and preventing data races or overlap between batches. This tightly coordinated mechanism enables efficient and orderly pipelined execution across model components, while keeping both memory footprint and runtime overhead to a minimum.

### A.3 ADDITIONAL EXPERIMENTAL RESULTS

#### A.3.1 IMPACT OF POLICY

We present detailed end-to-end throughput data, as shown in Table 6, Table 7, Table 8, Table 9, Table 10, and Table 11, to simulate different scenarios and analyze the impacts of policy on throughput by generating 16 tokens. Due to the large number of GPU hours required to complete all (prefill batch size, decoding batch size, draft batch size, draft max new tokens) combinations using Mixtral 8×22B in Env #2, we evaluated only a subset of possible configurations.

The prefill batch size is a tunable parameter for which the optimal value can be explicitly determined by the scheduling algorithm, as the Prefill stage does not involve speculative decoding and thus is free from probabilistic uncertainty. Under the experimental setup of Table 8, the optimal value is 80. For example, comparing entries 5 and 27 in Table 8—where all other parameters are held constant—the higher throughput of entry 27 is attributed to its more optimal Prefill batch size.

The decoding batch size and draft max new tokens jointly affect the verification latency of the target model. Since the target model's computation is offloaded to the CPU, the speculative decoding verification cannot achieve the same level of tight serialization as on the GPU. As a result, increasing the batch size or the number of new tokens leads to longer CPU computation time. As illustrated by entries 26–30 and 10, 20, 30, 40 in Table 8, neither a larger batch size nor a higher max new token value consistently yields better performance.

The decoding batch size, draft batch size, and draft max new tokens jointly impact the generation latency of the draft model. Due to GPU memory constraints, the draft batch size is typically limited to a small value. However, since all draft model computations are executed on the GPU, they are highly efficient. This allows the full decoding batch to be processed through a fine-grained, multi-round strategy. As long as the draft model's token generation time remains below the I/O-bound latency, it does not constitute a performance bottleneck.

The results in Table 11 exhibit a similar pattern. These four parameters are tightly coupled and collectively determine the overall system throughput. Given that our design introduces at least four tunable parameters, finding optimal settings through enumeration or heuristics alone is highly unlikely. This highlights the critical role of the Paraspec Planner in the overall system.

Table 6: Impact of policy on Mixtral 8x7B in Env #1, HumanEval dataset.

| No. | Prefill Batch Size | Decoding Batch Size | Draft batch size | Draft max new token | Throughput (token/s) |
|-----|-----|-----|-----|-----|-----|
| 1 | 80 | 200 | 10 | 8 | 32.821 |
| 2 | 80 | 160 | 6 | 1 | 15.869 |
| 3 | 80 | 160 | 6 | 2 | 20.964 |
| 4 | 80 | 160 | 6 | 4 | 28.914 |
| 5 | 80 | 160 | 6 | 6 | 33.711 |
| 6 | 80 | 160 | 6 | 8 | 33.690 |
| 7 | 80 | 160 | 8 | 1 | 15.834 |
| 8 | 80 | 160 | 8 | 2 | 20.940 |
| 9 | 80 | 160 | 8 | 4 | 29.267 |
| 10 | 80 | 160 | 8 | 6 | 32.520 |
| 11 | 80 | 160 | 8 | 8 | 32.776 |
| 12 | 80 | 160 | 10 | 1 | 15.835 |
| 13 | 80 | 160 | 10 | 2 | 21.120 |
| 14 | 80 | 160 | 10 | 4 | 29.499 |
| 15 | 80 | 160 | 10 | 6 | 32.226 |
| 16 | 80 | 160 | 10 | 8 | 32.540 |
| 17 | 80 | 200 | 6 | 1 | 18.736 |
| 18 | 80 | 200 | 6 | 2 | 24.737 |
| 19 | 80 | 200 | 6 | 4 | 29.091 |
| 20 | 80 | 200 | 6 | 6 | 31.641 |
| 21 | 80 | 200 | 6 | 8 | 33.014 |
| 22 | 80 | 200 | 8 | 1 | 18.828 |
| 23 | 80 | 200 | 8 | 2 | 24.813 |
| 24 | 80 | 200 | 8 | 4 | 30.452 |
| 25 | 80 | 200 | 8 | 6 | 32.649 |
| 26 | 80 | 200 | 8 | 8 | 31.884 |
| 27 | 80 | 200 | 10 | 1 | 18.865 |
| 28 | 80 | 200 | 10 | 2 | 24.675 |
| 29 | 80 | 200 | 10 | 4 | 30.363 |
| 30 | 80 | 200 | 10 | 6 | 32.716 |
| 31 | 80 | 200 | 10 | 8 | 33.072 |
| 32 | 80 | 256 | 6 | 1 | 21.166 |
| 33 | 80 | 256 | 6 | 2 | 26.052 |
| 34 | 80 | 256 | 6 | 4 | 30.279 |
| 35 | 80 | 256 | 6 | 6 | 32.325 |
| 36 | 80 | 256 | 6 | 8 | 32.812 |
| 37 | 80 | 256 | 8 | 1 | 20.683 |
| 38 | 80 | 256 | 8 | 2 | 27.123 |
| 39 | 80 | 256 | 8 | 4 | 31.829 |
| 40 | 80 | 256 | 8 | 6 | 33.622 |
| 41 | 80 | 256 | 8 | 8 | 33.247 |
| 42 | 80 | 256 | 10 | 1 | 20.546 |
| 43 | 80 | 256 | 10 | 2 | 26.987 |
| 44 | 80 | 256 | 10 | 4 | 30.679 |
| 45 | 80 | 256 | 10 | 6 | 34.665 |
| 46 | 80 | 256 | 10 | 8 | 33.445 |

Table 7: Impact of policy on Mixtral 8x7B in Env #1, C-Eval dataset.

| No. | Prefill Batch Size | Decoding Batch Size | Draft batch size | Draft max new token | Throughput (token/s) |
|---|---|---|---|---|---|
| 1 | 96 | 256 | 8 | 4 | **26.489** |
| 2 | 96 | 288 | 8 | 4 | **28.449** |
| 3 | 96 | 300 | 8 | 4 | **28.209** |
| 4 | 96 | 256 | 6 | 2 | **25.363** |
| 5 | 96 | 256 | 6 | 4 | **27.823** |
| 6 | 96 | 256 | 6 | 6 | **28.712** |
| 7 | 96 | 256 | 6 | 8 | **28.531** |
| 8 | 96 | 256 | 8 | 2 | **25.347** |
| 9 | 96 | 256 | 8 | 4 | **27.449** |
| 10 | 96 | 288 | 6 | 2 | **25.254** |
| 11 | 96 | 288 | 6 | 4 | **28.685** |
| 12 | 96 | 288 | 6 | 6 | **29.199** |
| 13 | 96 | 288 | 6 | 8 | **29.385** |
| 14 | 96 | 288 | 8 | 2 | **26.126** |
| 15 | 96 | 288 | 8 | 4 | **28.679** |
| 16 | 96 | 288 | 8 | 6 | **29.329** |
| 17 | 96 | 300 | 6 | 2 | **24.821** |
| 18 | 96 | 300 | 6 | 4 | **28.240** |
| 19 | 96 | 300 | 6 | 6 | **29.134** |
| 20 | 96 | 300 | 6 | 8 | **30.781** |
| 21 | 96 | 300 | 8 | 2 | **26.268** |
| 22 | 96 | 300 | 8 | 4 | **30.652** |
| 23 | 96 | 300 | 8 | 6 | **31.968** |

Table 8: Impact of policy on Mixtral 8x7B in Env #1, SummEval dataset.

| No. | Prefill Batch Size | Decoding Batch Size | Draft batch size | Draft max new token | Throughput (token/s) |
|---|---|---|---|---|---|
| 1 | 50 | 128 | 5 | 5 | 18.937 |
| 2 | 50 | 128 | 5 | 3 | 19.735 |
| 3 | 50 | 256 | 5 | 5 | 19.890 |
| 4 | 50 | 256 | 5 | 3 | 17.560 |
| 5 | 50 | 256 | 5 | 2 | 15.624 |
| 6 | 80 | 128 | 5 | 1 | 11.682 |
| 7 | 80 | 128 | 5 | 2 | 14.509 |
| 8 | 80 | 128 | 5 | 4 | 19.464 |
| 9 | 80 | 128 | 5 | 6 | 21.166 |
| 10 | 80 | 128 | 5 | 8 | 21.531 |
| 11 | 80 | 128 | 8 | 1 | 11.629 |
| 12 | 80 | 128 | 8 | 2 | 14.408 |
| 13 | 80 | 128 | 8 | 4 | 18.321 |
| 14 | 80 | 128 | 8 | 6 | 16.989 |
| 15 | 80 | 128 | 8 | 8 | 21.958 |
| 16 | 80 | 192 | 5 | 1 | 14.764 |
| 17 | 80 | 192 | 5 | 2 | 16.830 |
| 18 | 80 | 192 | 5 | 4 | 21.072 |
| 19 | 80 | 192 | 5 | 6 | 22.029 |
| 20 | 80 | 192 | 5 | 8 | 22.712 |
| 21 | 80 | 192 | 8 | 1 | 14.305 |
| 22 | 80 | 192 | 8 | 2 | 16.757 |
| 23 | 80 | 192 | 8 | 4 | 21.435 |
| 24 | 80 | 192 | 8 | 6 | 23.653 |
| 25 | 80 | 192 | 8 | 8 | 24.732 |
| 26 | 80 | 256 | 5 | 1 | 14.809 |
| 27 | 80 | 256 | 5 | 2 | 16.781 |
| 28 | 80 | 256 | 5 | 4 | 20.441 |
| 29 | 80 | 256 | 5 | 6 | 21.841 |
| 30 | 80 | 256 | 5 | 8 | 21.741 |
| 31 | 80 | 256 | 8 | 1 | 13.822 |
| 32 | 80 | 256 | 8 | 2 | 16.265 |
| 33 | 80 | 256 | 8 | 4 | 17.243 |
| 34 | 80 | 256 | 8 | 6 | 12.903 |
| 35 | 80 | 256 | 8 | 8 | 11.103 |
| 36 | 80 | 320 | 5 | 1 | 4.444 |
| 37 | 80 | 320 | 5 | 2 | 5.757 |
| 38 | 80 | 320 | 5 | 4 | 7.761 |
| 39 | 80 | 320 | 5 | 6 | 12.376 |
| 40 | 80 | 320 | 5 | 8 | 11.503 |
| 41 | 80 | 320 | 8 | 1 | 4.550 |
| 42 | 80 | 320 | 8 | 2 | 6.074 |
| 43 | 80 | 320 | 8 | 4 | 11.785 |
| 44 | 80 | 320 | 8 | 6 | 13.218 |
| 45 | 80 | 320 | 8 | 8 | 11.293 |

Table 9: Impact of policy on Mixtral 8x22B in Env #2, HumanEval dataset.

| No. | Prefill Batch Size | Decoding Batch Size | Draft batch size | Draft max new token | Throughput (token/s) |
|---|---|---|---|---|---|
| 1 | 32 | 128 | 4 | 4 | 7.112 |
| 2 | 32 | 128 | 4 | 6 | 7.921 |
| 3 | 32 | 128 | 4 | 8 | 7.564 |
| 4 | 32 | 128 | 6 | 4 | 8.617 |
| 5 | 32 | 128 | 6 | 6 | 7.901 |
| 6 | 32 | 128 | 6 | 8 | 7.155 |
| 7 | 32 | 128 | 8 | 4 | 6.599 |
| 8 | 32 | 128 | 8 | 6 | 7.913 |
| 9 | 32 | 128 | 8 | 8 | 7.677 |
| 10 | 32 | 192 | 4 | 4 | 7.291 |
| 11 | 32 | 192 | 4 | 6 | 7.083 |
| 12 | 32 | 192 | 4 | 8 | 4.874 |
| 13 | 32 | 192 | 6 | 4 | 7.753 |
| 14 | 32 | 192 | 6 | 4 | 7.733 |
| 15 | 32 | 192 | 6 | 6 | 7.578 |
| 16 | 32 | 192 | 6 | 8 | 4.510 |
| 17 | 32 | 192 | 8 | 4 | 8.536 |
| 18 | 32 | 192 | 8 | 6 | 6.574 |

Table 10: Impact of policy on Mixtral 8x22B in Env #2, C-Eval dataset.

| No. | Prefill Batch Size | Decoding Batch Size | Draft batch size | Draft max new token | Throughput (token/s) |
|---|---|---|---|---|---|
| 1 | 16 | 32 | 6 | 4 | 3.430 |
| 2 | 16 | 32 | 6 | 6 | 4.510 |
| 3 | 16 | 32 | 6 | 8 | 4.321 |
| 4 | 16 | 32 | 8 | 4 | 3.607 |
| 5 | 16 | 32 | 8 | 6 | 4.230 |
| 6 | 16 | 32 | 8 | 8 | 4.742 |
| 7 | 32 | 32 | 6 | 4 | 3.726 |
| 8 | 32 | 32 | 6 | 6 | 4.977 |
| 9 | 32 | 32 | 6 | 8 | 4.513 |
| 10 | 32 | 32 | 8 | 4 | 3.969 |
| 11 | 32 | 32 | 8 | 6 | 4.233 |
| 12 | 32 | 32 | 8 | 8 | 3.894 |
| 13 | 32 | 32 | 6 | 4 | 3.543 |
| 14 | 32 | 32 | 6 | 6 | 4.647 |
| 15 | 32 | 32 | 6 | 8 | 4.063 |
| 16 | 32 | 32 | 8 | 4 | 4.030 |
| 17 | 32 | 32 | 8 | 6 | 4.231 |
| 18 | 32 | 32 | 8 | 8 | 3.609 |
| 19 | 16 | 64 | 6 | 4 | 4.160 |
| 20 | 16 | 64 | 6 | 6 | 4.510 |
| 21 | 16 | 64 | 6 | 8 | 3.915 |
| 22 | 16 | 64 | 8 | 4 | 3.588 |

### A.3.2    ROBUSTNESS OF THE PARASPEC PLANNER

Table 12 below compares ParaSpec Policy to the empirically optimal policy; on average, ParaSpec policy achieves 93.7% of the optimal.

Table 11: Impact of policy on Mixtral 8x22B in Env #2, SummEval dataset.

| No. | Prefill Batch Size | Decoding Batch Size | Draft batch size | Draft max new token | Throughput (token/s) |
|-----|--------------------|---------------------|------------------|---------------------|----------------------|
| 1 | 16 | 64 | 6 | 4 | 4.579 |
| 2 | 16 | 32 | 6 | 4 | 3.711 |
| 3 | 16 | 32 | 6 | 6 | 3.486 |
| 4 | 16 | 32 | 6 | 8 | 4.225 |
| 5 | 16 | 32 | 8 | 4 | 3.862 |
| 6 | 16 | 32 | 8 | 6 | 3.998 |
| 7 | 16 | 32 | 8 | 8 | 3.975 |
| 8 | 16 | 64 | 6 | 4 | 4.529 |
| 9 | 16 | 64 | 6 | 6 | 5.141 |
| 10 | 16 | 64 | 6 | 8 | 4.977 |
| 11 | 16 | 64 | 8 | 4 | 4.546 |
| 12 | 16 | 64 | 8 | 6 | 4.590 |
| 13 | 16 | 64 | 8 | 8 | 5.911 |

Table 12: Gap between Optimal Policy Throughput and ParaSpec Policy Throughput (token/s). The policy is denoted as (prefill batch size, decoding batch size, draft batch size, draft max new tokens).

| Setup | Dataset | Optimal Throughput | ParaSpecThroughput | Efficiency |
|-------|---------|--------------------|--------------------|------------|
| 8x7B, Env #1 | Humaneval | 34.665 (80,256,10,6) | 33.072 (80,200,10,8) | 95% |
| 8x7B, Env #1 | C-Eval | 31.968 (96,300,8,6) | 30.781 (96,300,6,8) | 96% |
| 8x7B, Env #1 | SummEval | 24.743 (80,192,8,8) | 21.958 (80,128,8,8) | 89% |
| 8x22B, Env #2 | Humaneval | 8.617 (32,128,6,4) | 8.617 (32,128,6,4) | 100% |
| 8x22B, Env #2 | C-Eval | 4.977 (32,32,6,6) | 4.063 (32,32,6,8) | 82% |
| 8x22B, Env #2 | SummEval | 5.911 (16,64,8,8) | 5.911 (16,64,8,8) | 100% |

### A.3.3 EXTEND TO LARGE GPU VRAM

Table 13: Throughput (token/s) on different GPU VRAM for 8x7B, HumanEval, Env #1.

| VRAM (GB) | Throughput | Improvement |
|-----------|------------|-------------|
| 24 | 34.665 | / |
| 32 | 45.382 | 31% |
| 40 | 50.829 | 47% |
| 48 | 53.002 | 53% |

We added results for Mixtral 8x7B on an RTX 4090, where the available VRAM is increased from 24GB to 48GB. Despite the doubled memory, SpecOffload's throughput improved by around 53%. The results are in Table 13.

Increasing VRAM allocation yields limited benefit for both models: the draft model encounters diminishing returns due to an upper bound on token accuracy, and the target model also exhibits inefficient scaling as stated in the paper.

We hypothesize that assigning the extra memory to a more powerful draft model or adopting nonlinear draft token structures (e.g., Medusa (Cai et al., 2024), SpecInfer (Miao et al., 2024), or SpecExec (Svirschevski et al., 2024)) could improve this limitation. This is a promising direction for future work.

### A.3.4 DIFFERENT HARDWARE

We conducted experiments with SpecOffload across a range of hardware configurations, as shown in Table 14. The results demonstrate its strong practical applicability, showing that high inference throughput can be achieved consistently on both high-end and lower-end hardware.

Table 14: Throughput of SpecOffload in different hardware. Mixtral 8x7B, HumanEval.

| Hardware Configuration | Throughput (token/s) |
|---|---|
| Tesla T4 (16GB VRAM) PCIe 3.0, CPU Intel E5-2620 | 26.128 |
| RTX 4090 (24GB VRAM) PCIe 3.0, CPU Intel i9-10980XE | 34.665 |
| RTX 4090 (48GB VRAM) PCIe 3.0, CPU Intel i9-10980XE | 53.002 |
| RTX A6000 (48GB VRAM) PCIe 4.0, CPU Intel Platinum 8490H | 69.064 |

### A.3.5 GPU MEMORY USAGE

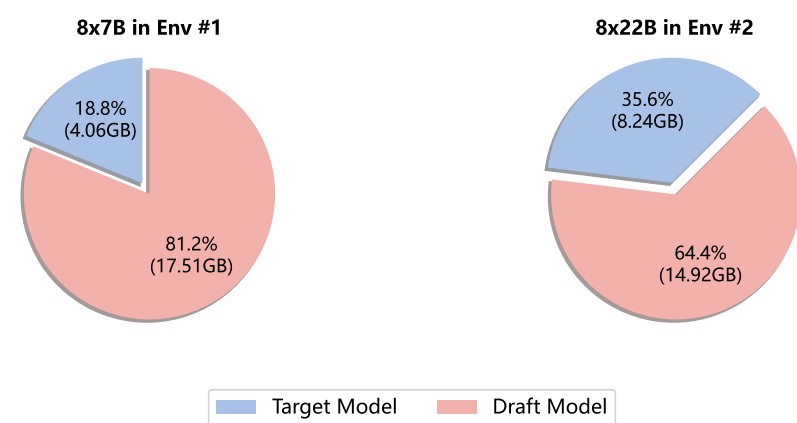

Figure 11: GPU Memory Allocation Overview.

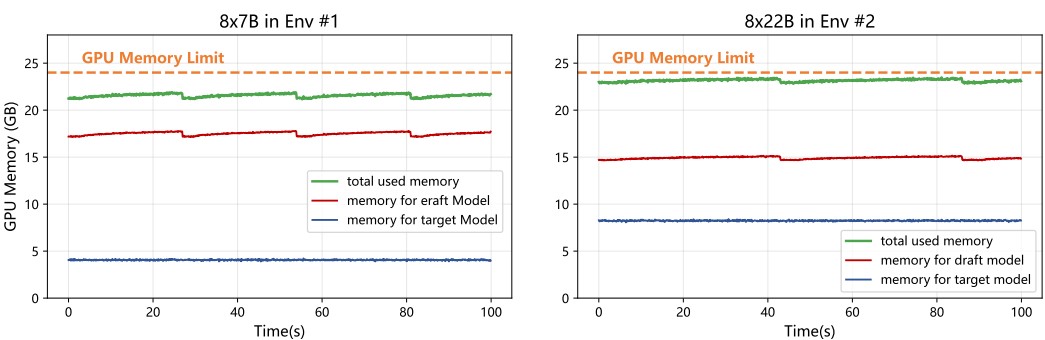

Figure 12: Runtime GPU Memory Monitoring.

We used NVIDIA Nsight nsi (2025) to monitor runtime GPU memory usage on the SummEval dataset. As shown in Figure 11, only the parameters essential for target model offloading are retained in memory, while the remaining space is occupied by the draft model and its cache. This aligns with our design rationale: during offloading, it is more efficient to allocate GPU memory to the draft model rather than storing the target model parameters.

Runtime GPU memory monitoring reveals a periodic pattern in the draft model's memory usage. As shown in the left panel of Figure 12, each cycle lasts approximately 28 seconds, characterized by a gradual increase in memory usage followed by a sharp drop and a 2-second idle window. This aligns with the behavior observed in Figure 6, where the draft model performs computation for 26 seconds and remains idle for 2 seconds awaiting the next batch.

A.3.6 ABLATION STUDY

In addition to the main results, we conducted ablation studies on other datasets. The results are as follows:

Table 15: Ablation study of proposed techniques on SummEval dataset. The gray tuple denotes a policy (prefill batch size, decoding batch size, draft batch size, draft max new tokens).

|  | All optimizations | No policy search | Serial SD | No SD |
|---|---|---|---|---|
| 8x7B | 24.743 (80, 192, 8, 8) | 15.624 (50, 256, 5, 2) | 17.048 (80, 192, 40, 8) | 12.369 (80, 256, x, x) |
| 8x22B | 5.911 (16, 64, 8, 8) | 3.486 (16, 32, 6, 6) | 4.146 (16, 64, 32, 8) | 1.698 (16, 80, x, x) |

Table 16: Ablation study of proposed techniques on C-Eval dataset. The gray tuple denotes a policy (Prefill batch size, decoding batch size, draft batch size, draft max new tokens).

|  | All optimizations | No policy search | Serial SD | No SD |
|---|---|---|---|---|
| 8x7B | 31.968 (96, 300, 8, 6) | 26.126 (96, 288, 8, 2) | 21.989 (96, 288, 24, 6) | 15.106 (96, 288, x, x) |
| 8x22B | 4.977 (32, 32, 6, 6) | 3.588 (16, 64, 8, 4) | 3.820 (32, 64, 16, 6) | 1.812 (32, 64, x, x) |

Table 17: Ablation study of proposed techniques on SAMSum dataset. The gray tuple denotes a policy (Prefill batch size, decoding batch size, draft batch size, draft max new tokens).

|  | All optimizations | No policy search | Serial SD | No SD |
|---|---|---|---|---|
| 8x7B | 21.109 (100, 300, 6, 4) | 12.694 (80, 256, 8, 2) | 13.64 (100, 300, 24, 4) | 13.072 (80, 256, x, x) |
| 8x22B | 4.139 (16, 64, 8, 6) | 3.059 (16, 64, 6, 4) | 3.544 (16, 64, 16, 6) | 2.378 (16, 80, x, x) |

A.4 EXTENSION TO DYNAMIC SERVING SCENARIOS

While the experiments in this paper primarily target single-user offline inference settings on resource-constrained edge devices, the core methodology of SpecOffload can be extended to dynamic or multi-user serving scenarios. This section discusses the necessary adaptations regarding the planner's overhead and the broader system pipeline.

Adapting the ParaSpec Planner for Real-Time Constraints. In the offline setting, the planner's runtime (approximately 5 seconds) is negligible compared to the total inference duration (hundreds of seconds). However, for dynamic scenarios where requests arrive stochastically, this overhead must be minimized. We propose two strategies to mitigate this: Search Space Reduction: By constraining the solution search space—for instance, reducing the grid dimensions from the original $100 \times 100 \times 20 \times 20$ to $50 \times 50 \times 5 \times 5$—the planner's execution time drops to <0.1s, making it feasible for online decision-making. Pipelined Execution: Even without reducing the search space, the planner's runtime is significantly shorter than the heavy model inference. Therefore, the planning process can be integrated into the pipeline asynchronously, executing within the "CPU bubbles" inherent to the SpecOffload mechanism, thereby effectively hiding the scheduling latency.

System-Level Extensions. Beyond the planner, enabling robust multi-user serving requires enhancements across all three stages of the SpecOffload pipeline: Input Stage: An admission controller is required to merge incoming requests into micro-batches while accounting for request length distribution, priority levels, and fairness constraints. Planner Stage: The ParaSpec Planner must be updated to incorporate dynamic workload characteristics, user-level latency targets (SLOs), and batch symmetry constraints to ensure efficient parallelization. Inference Stage: To prevent long-tail requests from blocking the pipeline, mechanisms such as continuous batching should be integrated, allowing the system to adapt batch sizes on the fly.

Limitations on Edge Hardware. It is important to note that SpecOffload is specifically optimized for large-batch, high-throughput execution on edge hardware. In resource-constrained environments,

multi-user concurrency is often limited by available compute and memory bandwidth. Furthermore, the reliance on large batches to maximize hardware utilization introduces inherent latency trade-offs, which may make dynamic serving less optimal compared to the single-user throughput-oriented setting designed for this work.

## A.5 LIMITATION

The main limitation of this paper lies in the fact that speculative decoding is not a consistently reliable method for acceleration. In extreme cases, none of the draft tokens in multiple batches may be accepted, which greatly limits the acceleration effect of SpecOffload.

