# OpenReview forum: "SpecOffload: Unlocking Latent GPU Capacity for LLM Inference on Resource-Constrained Devices"
_ICLR.cc/2026/Conference — Submitted to ICLR 2026_

### Official Review · Reviewer_F4bn · 2025-10-26

**Soundness:** 2
**Presentation:** 3
**Contribution:** 2
**Rating:** 4
**Confidence:** 4

**Summary:**

This paper proposes SpecOffload for memory-restrained LLM inference scenario, where the large model has to be offloaded to CPU or disk for restrained GPU memory. It introduces speculative decoding during offloading where GPU idles, which involves 3 main modifications: batch interleave of drafting and verificaition in decoding phase, adaptive tensor planner to optimize offloading i/o latency, and ParaSpec planner to maximize throughput with given configurations. It also utilizes CPU for attention computation of the base model for more efficient pipelining. Experiments show that SpecOffload improves GPU core utilization by 4.49× and boosts inference throughput by 2.36×.

**Strengths:**

The topic is highly related to practical inference scenario. LLM inference on resource-restrained hardware is an important direction.

The observation of ‘Marginal utility of GPU memory’ in the introduction is inspiring: When model size far exceeds GPU memory, if a portion of memory is further ‘restricted’ from being used, the marginal throughput degradation will be marginal because there have been too much offloading operations. This guarantees the effectiveness of inserting speculation into offloading stage, as the further occupied memory causes marginal degradation to the verification but enables speculation at GPU and hence boosts overall performances.

The demonstration figures (Fig.3, 4) are clear and informative, making the main design easily understood.

**Weaknesses:**

My main concern is about attention and FFN pipeline of the base model in sec4.1.2:

(1)	The paper said that, the attention computation is on CPU and FFN is on GPU, and the FFN is offloaded to GPU while attention being computed, forming a pipeline. However, the computation of attention and FFN has interleaved data dependencies: Attention Layer i+1 needs the result of FFN Layer i, and FFN Layer i+1 needs the result of Attention Layer i+1, etc.. Therefore, in my opinion, it is impossible to first run all the attention layers on CPU, then all FFN layers on GPU.

(2)	The attention computation on CPU is parallelized with the drafting on GPU. However, running attention computation of a much larger model on CPU would be much slower than running a full smaller model on GPU (in principle), so I think there would be a huge pipeline bubble bottlenecked by the CPU computation. This could limit the effectiveness of the proposed interleaved pipeline.

Other concerns:

The compared baselines are weak. SpecOffload shows significant acceleration compared to all the mentioned baselines, while they all involve no speculative decoding, only offloading optimizations. As a result, the speedup ratios may be attributed to adding speculative decoding, rather than the effectiveness of the proposed method itself.

The contribution is relatively incremental. The contribution is mainly an engineering integration of speculative decoding into offloading pipelines. It builds upon well-established components (speculative decoding, offload scheduling, pipeline batch scheduling), making the novelty relatively incremental.

**Questions:**

1. Could the authors provide detailed computation workflow, to clarify how the attention and FFN layers are parallelized?
2. How do the authors mitigate the potential pipeline bubble caused by performing attention computation on the CPU while the draft model runs on the GPU? Could they provide more detailed quantitative profiling data of each stage, to show the overall pipeline efficiency?
3. Since all compared baselines lack speculative decoding, how can we disentangle the gains from speculative decoding itself versus the proposed integration?

---

> ### Author Response · Authors · 2025-11-16
> **Author Response to Reviewer F4bn (1/2)**
>
> We appreciate the reviewer's valuable feedback on our pipeline and experiments, and we are grateful for the opportunity to enhance our paper.
>
> # 1. Clarification on pipeline
>
> > (1) The paper said that, the attention computation is on CPU and FFN is on GPU, and the FFN is offloaded to GPU while attention being computed, forming a pipeline. However, the computation of attention and FFN has interleaved data dependencies: Attention Layer i+1 needs the result of FFN Layer i, and FFN Layer i+1 needs the result of Attention Layer i+1, etc.. Therefore, in my opinion, it is impossible to first run all the attention layers on CPU, then all FFN layers on GPU.
> >
> > (2) The attention computation on CPU is parallelized with the drafting on GPU. However, running attention computation of a much larger model on CPU would be much slower than running a full smaller model on GPU (in principle), so I think there would be a huge pipeline bubble bottlenecked by the CPU computation. This could limit the effectiveness of the proposed interleaved pipeline.
>
> We apologize if the conciseness of Fig. 4 led to a misunderstanding.
>
> The pipeline on the right side of Fig. 4 illustrates that the target model computes layer-by-layer, while the draft model computes micro-batch by micro-batch.
>
> Fig. 4 illustrates a per-layer pipeline relative to the target model. In this pipeline, the Layer $i$ Attention computation (CPU) overlaps with the Layer $i$ FFN I/O (to GPU) and subsequent GPU computation. This process repeats for Layer $i+1$. For Mixtral 8x22B (56 layers), this entire pipeline repeats 56 times to complete one verification round. This is noted in the Fig. 4 caption: "Repeat all layers of the target model until the current verification round is completed."
>
> However, the Draft Model executes its full computation, though its macro-batch is partitioned into micro-batches (`draft_batch_size`) due to memory constraints. Fig. 4 shows the Draft Model running in these micro-batches concurrently.
>
> For example, consider a task with an input length of 512, a macro-batch size of 224, 5 draft tokens, and a Draft Model micro-batch size of 4. The Draft Model's *total task* is to compute 517 tokens (512 + 5). As we do not retain the KV cache for the Draft Model, it must recompute the full 517-token sequence (Prefill-to-Decoding) from scratch.
>
> In contrast, the Target Model executes only one FFN layer (decoding) per iteration. To process the entire macro-batch, this pipeline block executes 56 times (for 56 layers). Therefore, over these 56 iterations, the Target Model sequentially completes FFN layers 1 through 56. Concurrently, the Draft Model completes its *entire* Prefill-Decoding computation for the full batch, processing it cumulatively over 56 micro-batches (56 iterations $\times$ 4 micro-batch size = 224).
>
>
>
> > How do the authors mitigate the potential pipeline bubble caused by performing attention computation on the CPU while the draft model runs on the GPU? Could they provide more detailed quantitative profiling data of each stage, to show the overall pipeline efficiency?
>
> To further clarify, we provide quantitative data from Table 5:
>
> | Configuration        | Draft Model Computation (GPU) | Target Model Attention (CPU) |
> | -------------------- | ----------------------------- | ---------------------------- |
> | Mixtral 8x7B Env #1  | 489.02s                       | 531.23s                      |
> | Mixtral 8x22B Env #2 | 345.93s                       | 746.38s                      |
>
> Overall, these two computation times are on the same order of magnitude. Although the draft token length could be tuned to precisely align these durations, this is unnecessary. Our objective is to maximize throughput, not necessarily to minimize pipeline bubbles.

---

> ### Author Response · Authors · 2025-11-16
> **Author Response to Reviewer F4bn (2/2)**
>
> We appreciate the reviewer's valuable feedback on our pipeline and experiments, and we are grateful for the opportunity to enhance our paper.
>
> # 2. Disentangling the gains of pipeline
>
> > The compared baselines are weak. SpecOffload shows significant acceleration compared to all the mentioned baselines, while they all involve no speculative decoding, only offloading optimizations. As a result, the speedup ratios may be attributed to adding speculative decoding, rather than the effectiveness of the proposed method itself.
> > Since all compared baselines lack speculative decoding, how can we disentangle the gains from speculative decoding itself versus the proposed integration?
>
> We addressed this in our ablation study, with corresponding results in Table 4 、Table 15-17 of the paper. We evaluated variants of SpecOffload: one where speculative decoding is decoupled from the pipeline and executed serially ('Serial SD'), and one simulating the case where all speculations are incorrect ('No SD').
>
> We have reorganized these results below, as this experiment aims to disentangle the gains from speculative decoding itself versus the proposed integration.
>
> |                         | Dataset   | No SD | Serial SD | SpecOffload |
> | :---------------------- | :-------- | :---- | :-------- | :---------- |
> | Mixtral 8x7B Env #1  | HumanEval | 16.47 | 15.05     | 34.67       |
> | Mixtral 8x7B Env #1  | SummEval  | 12.37 | 17.05     | 24.74       |
> | Mixtral 8x7B Env #1  | C-Eval    | 15.11 | 21.99     | 31.97       |
> | Mixtral 8x7B Env #1  | SAMSum    | 13.07 | 13.64     | 21.11       |
> | Mixtral 8x22B Env #2 | HumanEval | 4.11  | 5.26      | 8.62        |
> | Mixtral 8x22B Env #2 | SummEval  | 1.70  | 4.15      | 5.91        |
> | Mixtral 8x22B Env #2 | C-Eval    | 1.81  | 3.82      | 4.98        |
> | Mixtral 8x22B Env #2 | SAMSum    | 2.38  | 3.54      | 4.14        |
>
> The results demonstrate that naively introducing SD does not guarantee performance gains, as the draft model introduces significant overhead from complex model switching and management. In contrast, only SpecOffload's pipeline architecture enables speculative decoding to be embedded into the offloading process at virtually zero cost.
>
> # 3. Clarification on novelty
> > The contribution is relatively incremental. The contribution is mainly an engineering integration of speculative decoding into offloading pipelines. It builds upon well-established components (speculative decoding, offload scheduling, pipeline batch scheduling), making the novelty relatively incremental.
>
> We believe our paper presents a novel contribution and appreciate the opportunity to reiterate core insights.
>
> First, we identify a counter-intuitive paradox in existing offloading frameworks: despite being designed for GPU constraints, they suffer from low GPU core utilization due to I/O bottlenecks, while large GPU memory usage offers limited performance gains. This profound diagnosis of "latent resources" is a key insight that has been overlooked by prior work.
>
> Second, we propose a new paradigm that embraces the I/O bottleneck rather than hiding it. By embedding speculative decoding into the offloading pipeline, we convert idle compute and underutilized memory during I/O waits into productive work for the draft model—unlocking latent GPU capacity with near-zero overhead.

---

> > ### Comment · Reviewer_F4bn · 2025-11-25
> >
> > Thank you for preparing the detailed rebuttal. I do think the contribution is relatively incremental. It is mainly about integrating speculative decoding into offloading idling time, which seems more related to engineering implementation rather than algorithmic creation. Furthermore, in my opinion the usage scenario is also quite limited and unrealistic: running a large model with huge batch size (often $\ge 100$) on resource-intense hardware.
> >
> > I will keep my score, but if all other reviewers agree to accept this paper, I am also OK with it.

---

> ### Author Response · Authors · 2025-11-25
> **Thank you for your time and consideration**
>
> We thank the reviewer for reading our rebuttal and providing these follow-up comments.
>
> **Clarification on novelty**. We would like to reiterate that our work uncovers the overlooked 'latent resources' hidden in I/O bottlenecks and introduces a strategy to harvest this idle capacity for speculative decoding, transforming bottlenecks into performance gains. This goes beyond simple engineering integration by fundamentally redefining how idle compute and memory are utilized in offloading pipelines. As noted in the rebuttal above.
>
> **Clarification on scenario**. We respectfully clarify that large-batch execution is a standard requirement for "back-of-house" offline tasks, distinct from real-time chat.
> * A vast number of LLM applications are "back-of-house" offline tasks where throughput (total completion time) is the only metric that matters. As mentioned in our paper, these include synthetic data generation, form processing, and data wrangling. A concrete example from our lab involves the local extraction of structured data from all financial invoices accumulated over the last three years. Constrained by privacy regulations to use local consumer-grade GPUs, we found that processing this multi-year archive sequentially was infeasible due to I/O overhead. Aggregating requests into large batches ($\ge 100$) proved to be the essential implementation strategy to effectively handle such a large-scale, offline workload.
> * Alignment with Academic Consensus. Our experimental setting aligns with a broad body of recent top-tier literature. These works all target throughput-oriented inference on memory-constrained GPUs and consistently adopt large batch sizes as the primary strategy to amortize I/O overhead and maximize hardware utilization. FlexGen (ICML 2023): Explicitly targets "high-throughput generative inference" on a single GPU, using large batch sizes to hide the latency of offloading weights. Klotski (ASPLOS 2025) : Similarly addresses the memory bottleneck by employing an expert-aware multi-batch pipeline, validating that aggregating requests into larger batches is essential for efficiency in this scenario. MoE-Lightning (ASPLOS 2025): Focuses on "High-throughput MoE inference on memory-constrained GPUs," likewise prioritizing throughput optimization under the exact same hardware constraints where large-batch processing is the standard norm.

---

### Official Review · Reviewer_2S5g · 2025-10-31

**Soundness:** 3
**Presentation:** 3
**Contribution:** 3
**Rating:** 6
**Confidence:** 4

**Summary:**

This paper addresses the inefficiency of LLM inference on single-GPU or resource-limited devices, where offloading model weights from GPU to CPU memory causes severe GPU under-utilization and minimal performance scaling with additional GPU memory. The authors propose SpecOffload, a system that integrates speculative decoding directly into the offloading pipeline to reclaim idle GPU compute and unused GPU memory.
The key idea is to use a lightweight draft model that runs during GPU idle periods (while waiting for CPU–GPU data transfer) and to store it in “low-yield” GPU memory that would otherwise be wasted. SpecOffload introduces three core techniques: (1) a dual-batch interleaved pipeline allowing draft and target models to run concurrently; (2) adaptive tensor placement across GPU/CPU/disk memory tiers; and (3) a ParaSpec planner that automatically tunes batch and token parameters under GPU memory constraints.
Experiments on Mixtral and LLaMA models show up to 4.5× GPU utilization and 2.36× throughput gains over strong offloading baselines (FlexGen, DeepSpeed-Inference, Fiddler), demonstrating strong practicality on consumer GPUs.

**Strengths:**

1. The paper identifies an under-explored inefficiency — GPU idleness during offloading — and proposes to exploit this latent capacity through speculative decoding.
2. The dual-batch interleaved execution and adaptive tensor placement form a coherent system that effectively overlaps CPU compute, GPU compute, and I/O, where speculative decoding is repurposed not merely as a speed-up technique but as a way to hide I/O latency, which is novel in the offloading context.
3. The paper is well-written for a systems paper.

**Weaknesses:**

1. The speculative decoding itself and its integration into offloading is not unique. That being said, the novelty lies primarily in system integration and scheduling.
2. All experiments are single-GPU; no exploration of distributed or multi-GPU extension.

**Questions:**

1. The ParaSpec Planner seems tuned for specific hardware (RTX 4090). Could the authors elaborate on how much manual profiling or calibration is needed to port SpecOffload to a different GPU architecture.
2. In practice, how should users choose the draft model size relative to the target model? Nowadays many choose to use Eagle to reduce the spec execution time but it might not be too much helpful for an offloading system like SpecOffload. How should SpecOffload adapt in terms of scheduling policy?

---

> ### Author Response · Authors · 2025-11-16
> **Author Response to Reviewer 2S5g (1/2)**
>
> We appreciate the reviewer's positive feedback on our system design, and acknowledge their concerns regarding its novelty and specific module usage.
>
>
>
> # 1. Clarification on novelty
>
> > The speculative decoding itself and its integration into offloading is not unique. That being said, the novelty lies primarily in system integration and scheduling.
>
> We believe our paper presents a novel contribution and appreciate the opportunity to reiterate core insights.
>
> First, we identify a counter-intuitive paradox in existing offloading frameworks: despite being designed for GPU constraints, they suffer from low GPU core utilization due to I/O bottlenecks, while large GPU memory usage offers limited performance gains. This profound diagnosis of "latent resources" is a key insight that has been overlooked by prior work.
>
> Second, we propose a new paradigm that embraces the I/O bottleneck rather than hiding it. By embedding speculative decoding into the offloading pipeline, we convert idle compute and underutilized memory during I/O waits into productive work for the draft model—unlocking latent GPU capacity with near-zero overhead.
>
>
>
> # 2. Clarification on multi-GPU extension
>
> > All experiments are single-GPU; no exploration of distributed or multi-GPU extension.
>
> Yes, all experiments were conducted on a single GPU. This aligns with our paper's defined scope, as stated in the Introduction: "Our focus is on designing efficient offloading strategies for high-throughput inference on resource constrained devices with a single GPU."
>
> * This single-GPU setup is well-justified, as multi-GPU configurations are uncommon in such target scenarios. For instance, popular tools like JetBrains AI and LM Studio are designed for single-GPU PCs. This focus is also consistent with related academic work, including Klotski [1], Fiddler [2], and HeteGen [3], which all conducted experiments in single-GPU settings.
> * Theoretically, SpecOffload can be extended to multi-GPU setups, where additional GPUs could provide dedicated compute and memory resources for both the target and draft models.
>
> The extension of SpecOffload to multi-GPU configurations is planned as future work.
>
> [1]: Fang, Zhiyuan, et al. "Klotski: Efficient Mixture-of-Expert Inference via Expert-Aware Multi-Batch Pipeline." Proceedings of the ACM International Conference on Architectural Support for Programming Languages and Operating Systems. ASPLOS 2025.
>
> [2]: Kamahori, Keisuke, et al. "Fiddler: Cpu-gpu orchestration for fast inference of mixture-of-experts models." International Conference on Learning Representations. ICLR 2025.
>
> [3]: Zhao, Xuanlei, et al. "Hetegen: Efficient heterogeneous parallel inference for large language models on resource-constrained devices." Proceedings of Machine Learning and Systems. MLSys 2024.

---

> ### Author Response · Authors · 2025-11-16
> **Author Response to Reviewer 2S5g (2/2)**
>
> We appreciate the reviewer's positive feedback on our system design, and acknowledge their concerns regarding its novelty and specific module usage.
>
> # 3. Clarification on the planner's adaptability
>
> > The ParaSpec Planner seems tuned for specific hardware (RTX 4090). Could the authors elaborate on how much manual profiling or calibration is needed to port SpecOffload to a different GPU architecture.
>
> Extending the Planner to new CPU-GPU and model combinations is a two-step process that takes less than 5 minutes.
>
> * First, a profiler script measures the CPU/GPU execution latencies of model modules and the device's PCIe bandwidth. We leverage the open-source work from Fiddler for this profiling.
> * Second, this profiled data is input into the Planner script and executed.
>
> We conducted performance experiments across 4 hardware configurations (including 2 non-4090 GPUs), 3 model configurations, and 4 datasets. The Planner finds a solution in under 10 seconds, and this solution yields 93% of the optimal throughput.
>
> | Hardware                         | Model                        | Dataset   | Runtime(s) | Efficiency |
> | :------------------------------- | :--------------------------- | :-------- | :--------- | :--------- |
> | Env #1                           | Mistral 7B & Mixtral 8x7B    | HumanEval | 4.37       | 95%        |
> | Env #1                           | Mistral 7B & Mixtral 8x7B    | C-Eval    | 4.48       | 96%        |
> | Env #1                           | Mistral 7B & Mixtral 8x7B    | SummEval  | 3.50       | 89%        |
> | Env #1                           | Mistral 7B & Mixtral 8x7B    | SAMSum    | 4.42       | 92%        |
> | Env #2                           | Mistral 7B & Mixtral 8x22B   | HumanEval | 5.20       | 100%       |
> | Env #2                           | Mistral 7B & Mixtral 8x22B   | C-Eval    | 5.29       | 82%        |
> | Env #2                           | Mistral 7B & Mixtral 8x22B   | SummEval  | 4.04       | 100%       |
> | Env #2                           | Mistral 7B & Mixtral 8x22B   | SAMSum    | 5.10       | 93%        |
> | Env #1                           | LLaMA-3.1-8B & LLaMA-3.3-70B | HumanEval | 4.62       | 91%        |
> | Env #1                           | LLaMA-3.1-8B & LLaMA-3.3-70B | C-Eval    | 4.31       | 94%        |
> | Env #1                           | LLaMA-3.1-8B & LLaMA-3.3-70B | SummEval  | 3.42       | 88%        |
> | Env #1                           | LLaMA-3.1-8B & LLaMA-3.3-70B | SAMSum    | 4.48       | 93%        |
> | Intel E5-2620 + Tesla T4         | Mistral 7B & Mixtral 8x7B    | HumanEval | 8.87       | 96%        |
> | Intel Platinum 8490H + RTX A6000 | Mistral 7B & Mixtral 8x7B    | HumanEval | 3.12       | 87%        |
>
>
>
> # 4. On draft model selection and synergy with Eagle
>
> > In practice, how should users choose the draft model size relative to the target model? Nowadays many choose to use Eagle to reduce the spec execution time but it might not be too much helpful for an offloading system like SpecOffload. How should SpecOffload adapt in terms of scheduling policy?
>
> In practice, we select the largest compatible model from the same family as the draft model, reserving ~2GB for the KV cache.
>
> Indeed, numerous speculative decoding works, such as Eagle, exist. These efforts generally follow two directions: (1) Generating more accurate drafts; e.g., Eagle predicts features and incorporates Shifted-token correction. (2) Employing non-linear draft structures; e.g., Eagle generates multi-branch draft tokens in a single forward pass. The objective of these approaches is to increase the number of accepted draft tokens per iteration.
>
>  Eagle and related works are orthogonal to SpecOffload and can be integrated with it, as their speculative generation components can serve as more potent draft model substitutes.
>
> The reduced speculative execution time afforded by SpecOffload would provide more computational headroom, enabling the adoption of more complex and longer draft token structures to increase the accepted token length. This constitutes a direction for our future work.

---

### Official Review · Reviewer_7n7D · 2025-11-01

**Soundness:** 2
**Presentation:** 2
**Contribution:** 2
**Rating:** 4
**Confidence:** 4

**Summary:**

The paper makes an observation that there are underutilization of GPU cores due to the memory communication between CPU and GPU. Also, the paper claims that adding more GPU has a minimal effect on overall throughput as the model parameters are too large to reside in GPU memory permanently. introduces SpecOffload which is aims to utilize the idle GPU time to run lightweight draft model for speculative decoding. It claims that storing the draft model in the GPU memory would have negligible impact on performance. The paper uses a dual-batch interleaved design enables the target model and draft model to run concurrently, increasing GPU resource utilization.

**Strengths:**

* Efficient LLM inference is important considering its increasing adoption in many applications. As such the paper is working on a good direction.
* The paper aims to better utilize GPUs which is important and seems to get some gains.
* Amalgamation of Offloading and Speculative Decoding seems to be a clear nice followup for the Offloading works and Speculative Decoding works.

**Weaknesses:**

* The paper seems to be rather shallow on the experiment depth (please refer to the questions for details). This significantly limits its generalizability.

**Questions:**

* If the draft model is really small. How does this compare to the works that tries to leverage CPU resources for some part of the LLM compute. How would this perform if CPU's AMX units are used for draft model inference and GPU for the target model inference?
    * Na, Seonjin, et al. "FlexInfer: Flexible LLM Inference with CPU Computations." Eighth Conference on Machine Learning and Systems. 2025
* The paper mentions that speculative decoding is not always reliable and that its acceleration effect is limited if none of the draft tokens are accepted. However, it does not provide an in-depth analysis of the draft token acceptance rate, which is a critical factor for the system's performance. Can the authors add more experimental data on how this rate varies across different models, datasets, or hardware environments?
* While the paper states the planner achieves 93.7% of optimal policy performance on average, it does not provide detailed information on its computational cost, runtime overhead, or how it performs in more varied or complex scenarios. Can you provide more details? Also, can you provide more details as to how this can be generalized to different CPU-GPU combinations + models?
* It seems as the work is only focused on increasing the throughput in single-user scenario. How would this be extended to multi-user / dynamic serving scenarios?
* A.3.3 Table 13 seems to suggest that there is only 53% improvement for even double VRAM. Could you elaborate. It would also help if utilization data are provided?

---

> ### Author Response · Authors · 2025-11-17
> **Author Response to Reviewer 7n7D (1/3)**
>
> We thank the reviewers for recognizing the importance and novelty of our work, and we are grateful for the opportunity to enhance our paper.
>
> ## 1. Comparison to CPU-based draft model inference
>
> > If the draft model is really small. How does this compare to the works that tries to leverage CPU resources for some part of the LLM compute. How would this perform if CPU's AMX units are used for draft model inference and GPU for the target model inference?
>
> As shown in Fig. 4 (right), our inference pipeline already includes CPU-side attention computation (green blocks), following a widely adopted design in prior systems such as *FlexGen*, *HeteGen*, *MoE-Lightning*, and *Fiddler*. We have compared against the open-source implementations of *FlexGen* and *Fiddler*, and the results are reported in Fig. 5.
>
> In principle, both CPU and GPU execution of the draft model are possible as long as the throughput matches the pipeline schedule. However, CPU and GPU compute capabilities differ by orders of magnitude. Even for very small models, CPU throughput is insufficient to sustain a bubble-free pipeline.
>
> Our empirical results confirm this: a Mistral-7B draft model yields 423.4 tokens/s on GPU, but only 6.395 tokens/s on CPU. Even the Llama-3.2-1B model reaches only 18.358 tokens/s on CPU. Therefore, placing the draft model on CPU would introduce substantial pipeline stalls, making GPU-based draft execution the only viable design for maintaining throughput.
>
> ## 2. Analysis of draft token acceptance rate
>
> > The paper mentions that speculative decoding is not always reliable and that its acceleration effect is limited if none of the draft tokens are accepted. However, it does not provide an in-depth analysis of the draft token acceptance rate, which is a critical factor for the system's performance. Can the authors add more experimental data on how this rate varies across different models, datasets, or hardware environments?
>
> We thank the reviewer for highlighting the importance of draft token acceptance rate. We have added experiments covering multiple models and datasets, summarized in the table below. τ denotes the average number of tokens accepted per iteration, and a larger τ corresponds to a higher token acceptance rate. Across settings, we observe that higher acceptance rates consistently yield higher throughput.
>
> | Model            | Case 1                  | Case 2                  | Case 3                  |
> | ---------------- | ----------------------- | ----------------------- | ----------------------- |
> | openai_humaneval | τ=0.8,  20.214 tokens/s | τ=1.4,  23.425 tokens/s | τ=2.0,  26.372 tokens/s |
> | summeval         | τ=0.7,  14.355 tokens/s | τ=1.2,  17.714 tokens/s | τ=1.7,  20.087 tokens/s |
> | samsum           | τ=0.6,  16.859 tokens/s | τ=1.0,  20.079 tokens/s | τ=1.4,  21.486 tokens/s |
> | ceval_exam       | τ=0.8,  23.111 tokens/s | τ=1.4,  23.620 tokens/s | τ=1.9,  25.965 tokens/s |
>
> We additionally include an ablation examining an extreme case: when no draft token is accepted (No SD), speculative decoding provides no speedup. Even under this worst-case scenario, our system’s throughput remains comparable to the best baseline (*FlexGen*). This demonstrates that speculative decoding introduces near-zero overhead and does not degrade pipeline throughput even when unsuccessful.
>
> |                      | DataSet   | FlexGen | No SD |
> | -------------------- | --------- | ------- | ----- |
> | Mixtral 8x7B Env #1  | HumanEval | 15.06   | 16.47 |
> | Mixtral 8x7B Env #1  | C-Eval    | 14.93   | 15.11 |
> | Mixtral 8x7B Env #1  | SummEval  | 9.77    | 12.37 |
> | Mixtral 8x7B Env #1  | SAMSum    | 8.20    | 13.07 |
> | Mixtral 8x22B Env #2 | HumanEval | 4.38    | 4.11  |
> | Mixtral 8x22B Env #2 | C-Eval    | 2.11    | 1.81  |
> | Mixtral 8x22B Env #2 | SummEval  | 2.45    | 1.70  |
> | Mixtral 8x22B Env #2 | SAMSum    | 2.12    | 2.38  |

---

> > ### Comment · Reviewer_7n7D · 2025-11-24
> > **Question regarding CPU experiments**
> >
> > Thanks for the response.
> >
> > > In principle, both CPU and GPU execution of the draft model are possible as long as the throughput matches the pipeline schedule. However, CPU and GPU compute capabilities differ by orders of magnitude. Even for very small models, CPU throughput is insufficient to sustain a bubble-free pipeline.
> > >
> > > Our empirical results confirm this: a Mistral-7B draft model yields 423.4 tokens/s on GPU, but only 6.395 tokens/s on CPU. Even the Llama-3.2-1B model reaches only 18.358 tokens/s on CPU. Therefore, placing the draft model on CPU would introduce substantial pipeline stalls, making GPU-based draft execution the only viable design for maintaining throughput.
> >
> > It seems that the authors assume no AMX / AVX for CPU numbers. Recent research on LLM inference on CPUs suggest that using AMX (matrix extensions on Intel CPU) helps improve the throughput. While the author's claims are true for large models, it seems that the authors' response are completely neglecting the possibility of this alternative.

---

> > > ### Author Response · Authors · 2025-11-24
> > > **Response to CPU experiments**
> > >
> > > We thank the reviewer.
> > >
> > > * The CPU experimental data we provided was obtained with the AVX-512 instruction set enabled.
> > >
> > > * Due to the lack of access to machines supporting the AMX instruction set, we refer to online resources. The blog "Intel AMX Enables High-Efficiency CPU Inference for AI Workloads" indicates that the throughput of the **INT8-quantized** DeepSeek-R1 is 23 tokens/s, and the throughput of the **INT8-quantized** Llama3.2-3B is 53 tokens/s. In comparison, the **unquantized** Mistral-7B model achieves a throughput of 423.4 tokens/s on a 4090 GPU. The gap between them remains significant.
> > >
> > > Therefore, our conclusion remains unchanged: Placing the draft model on CPU would introduce substantial pipeline stalls, making GPU-based draft execution the only viable design for maintaining throughput.

---

> ### Author Response · Authors · 2025-11-17
> **Author Response to Reviewer 7n7D (2/3)**
>
> We thank the reviewers for recognizing the importance and novelty of our work, and we are grateful for the opportunity to enhance our paper.
>
> ## 3. Clarification of the planner
>
> > While the paper states the planner achieves 93.7% of optimal policy performance on average, it does not provide detailed information on its computational cost, runtime overhead, or how it performs in more varied or complex scenarios. Can you provide more details? Also, can you provide more details as to how this can be generalized to different CPU-GPU combinations + models?
>
> Our planner is CPU-only and typically completes within 10 seconds under standard hardware configurations, while achieving 93% of the optimal throughput. We conducted systematic evaluations on 4 hardware setups, 3 model configurations, and 4 datasets. The planner explores a 100 × 100 × 20 × 20 search space over $(bs_{\text{prefill}}, bs, bs_{\text{draft}}, n_{\text{cand}})$, with results presented in the table below.
>
> | Hardware                         | Model                        | Dataset   | Runtime(s) | Efficiency |
> | -------------------------------- | ---------------------------- | --------- | ---------- | ---------- |
> | Env #1                           | Mistral 7B & Mixtral 8x7B    | HumanEval | 4.37       | 95%        |
> | Env #1                           | Mistral 7B & Mixtral 8x7B    | C-Eval    | 4.48       | 96%        |
> | Env #1                           | Mistral 7B & Mixtral 8x7B    | SummEval  | 3.50       | 89%        |
> | Env #1                           | Mistral 7B & Mixtral 8x7B    | SAMSum    | 4.42       | 92%        |
> | Env #2                           | Mistral 7B & Mixtral 8x22B   | HumanEval | 5.20       | 100%       |
> | Env #2                           | Mistral 7B & Mixtral 8x22B   | C-Eval    | 5.29       | 82%        |
> | Env #2                           | Mistral 7B & Mixtral 8x22B   | SummEval  | 4.04       | 100%       |
> | Env #2                           | Mistral 7B & Mixtral 8x22B   | SAMSum    | 5.10       | 93%        |
> | Env #1                           | LLaMA-3.1-8B & LLaMA-3.3-70B | HumanEval | 4.62       | 91%        |
> | Env #1                           | LLaMA-3.1-8B & LLaMA-3.3-70B | C-Eval    | 4.31       | 94%        |
> | Env #1                           | LLaMA-3.1-8B & LLaMA-3.3-70B | SummEval  | 3.42       | 88%        |
> | Env #1                           | LLaMA-3.1-8B & LLaMA-3.3-70B | SAMSum    | 4.48       | 93%        |
> | Intel E5-2620 + Tesla T4         | Mistral 7B & Mixtral 8x7B    | HumanEval | 8.87       | 96%        |
> | Intel Platinum 8490H + RTX A6000 | Mistral 7B & Mixtral 8x7B    | HumanEval | 3.12       | 87%        |
>
> Generalizing the planner to new CPU–GPU combinations or new models involves only two lightweight steps (≤5 minutes total):
>
> 1. Profiling.
>
>    A provided profiler (built upon Fiddler’s open-source code) measures individual CPU/GPU module runtimes and PCIe bandwidth on the target device.
>
> 2. Planning.
>    The measured values are fed into the planner script, which then automatically produces the scheduling policy.
>
> This design ensures that the planner generalizes efficiently to heterogeneous hardware and arbitrary model choices.
>
> ## 4. Extension to dynamic serving scenarios
>
> > It seems as the work is only focused on increasing the throughput in single-user scenario. How would this be extended to multi-user / dynamic serving scenarios?
>
> We agree with the reviewer that our experiments target the single-user offline inference setting. While SpecOffload is primarily designed for resource-constrained edge devices, its core ideas can be extended to multi-user or dynamic serving scenarios. Such an extension requires enhancements to all three stages of the pipeline:
>
> - Input stage: merge requests into micro-batches while accounting for request length, priority, and fairness.
> - ParaSpec Planner: incorporate dynamic workloads, user-level latency targets, and batch symmetry constraints.
> - Inference stage: integrate continuous batching or similar mechanisms to adapt batch sizes on the fly and prevent long-tail requests from blocking the pipeline.
>
> Nevertheless, we emphasize that SpecOffload is optimized for large-batch, high-throughput pipelines on edge hardware, where multi-user concurrency is often infeasible due to limited compute. The reliance on large batches also introduces latency trade-offs that make dynamic serving less ideal for our setting.

---

> > ### Comment · Reviewer_7n7D · 2025-11-24
> > **Question about extension to dynamic serving scenarios**
> >
> > Thanks for the detailed response to the questions.
> >
> > > Our planner is CPU-only and typically completes within 10 seconds under standard hardware configurations,
> >
> > > While SpecOffload is primarily designed for resource-constrained edge devices, its core ideas can be extended to multi-user or dynamic serving scenarios.
> >
> > It seems that the authors suggest that the ideas can be extended to dynamic scenarios. While most of the ideas may be extended, it seems that the 10 seconds is a big overhead that could significantly limit the performance of the idea in dynamic setting. It would be great if the discussion could be added to the paper regarding how this can be mitigated or optimized to be extended to dynamic setting. Possibly in appendix.

---

> ### Author Response · Authors · 2025-11-17
> **Author Response to Reviewer 7n7D (3/3)**
>
> We thank the reviewers for recognizing the importance and novelty of our work, and we are grateful for the opportunity to enhance our paper.
>
> ## 5. Clarification on Table 13
>
> > A.3.3 Table 13 seems to suggest that there is only 53% improvement for even double VRAM. Could you elaborate. It would also help if utilization data are provided?
>
> In the experiment corresponding to Table 13, we increase the GPU VRAM usage by adjusting the configuration planned by our scheduler. Specifically, we enlarge the prefill batch size $bs_{\text{prefill}}$, the decoding batch size $bs$, the draft model inference batch size $bs_{\text{draft}}$, and the number of draft tokens $n_{\text{cand}}$, which collectively improves throughput. We also provide the GPU utilization statistics below:
>
> | VRAM(GB) | Throughput(token/s) | GPU utilization |
> | -------- | ------------------- | --------------- |
> | 24       | 34.67               | 58.67%          |
> | 32       | 45.38               | 60.22%          |
> | 40       | 50.83               | 60.31%          |
> | 48       | 53.00               | 60.91%          |
>
> However, such parameter scaling is not an effective strategy for improving VRAM utilization. In practice, when more VRAM is available, SpecOffload should leverage larger draft models and more expressive non-linear draft token structures so that the additional memory is converted into higher draft-token acceptance rates, thereby maximizing throughput.

---

> ### Author Response · Authors · 2025-11-24
> **Response about extension to dynamic serving scenarios**
>
> We thank the reviewer for the insight. While the planner's 10s overhead is negligible in the target offline scenario (compared to hundreds of seconds of inference), we acknowledge its impact in dynamic scenarios. We can address this via two approaches:
>
> 1.  **Reduce the search space:** Constraining the solution space from the original $100 \times 100 \times 20 \times 20$ to **$50 \times 50 \times 5 \times 5$** drops the runtime to <0.1s, making the overhead acceptable.
>
> 2.  **Pipeline integration:** Since planning is significantly faster than inference, it can be executed within the "CPU bubbles" of SpecOffload, effectively hiding the latency without reducing the search space.
>
> We have included this discussion in Appendix A.4 of the revised PDF.

---

### Official Review · Reviewer_D7ni · 2025-11-04

**Soundness:** 3
**Presentation:** 2
**Contribution:** 3
**Rating:** 6
**Confidence:** 3

**Summary:**

This paper proposes an LLM inference system that combines offloading and speculative decoding. It executes the draft model concurrently on the idle GPU with the offloading and CPU computation of the target model. It also provides modeling of the execution pipeline and planner to search for the configuration. Evaluation show that this work has superior performance than the related offloading-based LLM inference systems.

**Strengths:**

1. This paper focuses on an important topic of efficient LLM serving.
2. It proposes a clever method to fill the draft model execution into the bubbles of offloading of the target model.

**Weaknesses:**

1. The planner part is not clear enough, nor necessary enough.
2. Some analysis and claim are not solid enough.

**Questions:**

1. Why put the Attention computation of the target model on the CPU? Is it only to save the GPU memory for the KV cache? Note Attention is usually memory bound and the GPU has higher bandwidth than the CPU.
2. The results in Figure 2 lack of some details. How is the increased GPU memory used? For example, one method can be making some portion of the weight persistent in the increased GPU memory and thus the performance can improve gradually.
3. In Figure 4, why the execution of the Draft model is longer than the FFN of the target model? (Note for moderate sequence length, the execution of the target model on the GPU should be dominated by its FFN.)
4. In Section 4.3, how the equation 1 is solved? Besides, equation 4 seems not accurate. For example, if $T_{draft}$ is very long, then the FFN of the target model will need to wait for the end of the draft model (according to Figure 4). In this case, the time will not be the max of the two, but seems $T_{draft} + T_{target-ffn}$.
5. It is not clear what is the search space of the planner. It seems everything is deterministic, or can be defined very easy. So the planner seems not necessary.

---

> ### Author Response · Authors · 2025-11-16
> **Author Response to Reviewer D7ni (1/2)**
>
> We thank the reviewer for their valuable suggestions regarding the pipeline and the planner, and we are pleased to address them here.
>
> # 1. On the design choice of CPU attention
>
> > Why put the Attention computation of the target model on the CPU? Is it only to save the GPU memory for the KV cache? Note Attention is usually memory bound and the GPU has higher bandwidth than the CPU.
>
> We execute decoding-phase Attention on the CPU to maximize throughput, for two key reasons:
>
> * It enables larger batches. Keeping the KV cache in CPU memory supports larger batch sizes, which is essential to amortize the I/O cost of loading FFN parameters.
>
> * It is faster in an offloading context. Although Attention is memory-bound and theoretically suited for the GPU, the I/O overhead far exceeds the computation time. For example (on Env#1, Mixtral 8x7B, input=512), direct CPU computation (2.6 ms) is significantly faster than the alternative of transferring the KV cache back to the GPU (5.0 ms I/O) *plus* the GPU computation (0.7 ms).
>
> While this design also saves GPU memory, its primary purpose is maximizing throughput.
>
> # 2. Clarification on Fig. 2
>
> > The results in Figure 2 lack of some details. How is the increased GPU memory used? For example, one method can be making some portion of the weight persistent in the increased GPU memory and thus the performance can improve gradually.
>
> In the approach illustrated in Fig. 2, while additional GPU memory is used to pin more FFN layers, the resulting performance gains are minimal.
>
> This figure uses FlexGen's strategy of CPU-Attention and GPU-FFN. Take Mixtral-8x22B as an example. The leftmost point uses ~7 GB VRAM and pins 0 FFN layers. The rightmost point uses ~21 GB VRAM but pins only 3 FFN layers.
>
> Due to autoregressive decoding, the model must still load all 56 FFN layers sequentially to generate one token. The two setups must therefore load 56 versus 53 layers. This reduces I/O time by only 5%.
>
> In offloading, I/O latency dominates computation, so total latency is determined by data loading. This is why, despite more VRAM, the overall latency and throughput for both setups remain nearly identical.
>
> We discuss this point in the paper in lines 82-92.
>
> # 3. Clarification on pipeline
>
> > In Figure 4, why the execution of the Draft model is longer than the FFN of the target model? (Note for moderate sequence length, the execution of the target model on the GPU should be dominated by its FFN.)
>
> We apologize if the conciseness of Fig. 4 led to a misunderstanding.
>
> The pipeline on the right side of Fig. 4 illustrates that the target model computes layer-by-layer, while the draft model computes micro-batch by micro-batch. herefore, it is reasonable and expected that the draft model's execution time per block is significantly longer, and we apologize for the confusion.
>
> Fig. 4 illustrates a per-layer pipeline relative to the target model. In this pipeline, the Layer $i$ Attention computation (CPU) overlaps with the Layer $i$ FFN I/O (to GPU) and subsequent GPU computation. This process repeats for Layer $i+1$. For Mixtral 8x22B (56 layers), this entire pipeline repeats 56 times to complete one verification round. This is noted in the Fig. 4 caption: "Repeat all layers of the target model until the current verification round is completed."
>
> However, the Draft Model executes its full computation, though its macro-batch is partitioned into micro-batches (`draft_batch_size`) due to memory constraints. Fig. 4 shows the Draft Model running in these micro-batches concurrently.
>
> For example, consider a task with an input length of 512, a macro-batch size of 224, 5 draft tokens, and a Draft Model micro-batch size of 4. The Draft Model's *total task* is to compute 517 tokens (512 + 5). As we do not retain the KV cache for the Draft Model, it must recompute the full 517-token sequence (Prefill-to-Decoding) from scratch.

---

> ### Author Response · Authors · 2025-11-16
> **Author Response to Reviewer D7ni (2/2)**
>
> We thank the reviewer for their valuable suggestions regarding the pipeline and the planner, and we are pleased to address them here.
>
> # 4. Clarification on Eq.1 and Eq.4
> > In Section 4.3, how the equation 1 is solved? Besides, equation 4 seems not accurate. For example, if is very long, then the FFN of the target model will need to wait for the end of the draft model (according to Figure 4). In this case, the time will not be the max of the two, but seems  $T_{draft}+T_{target-ffn}$
>
> We formulate throughput maximization as an optimization problem. The objective is to maximize throughput, subject to the constraint of GPU memory capacity. The search space comprises the inference parameters of SpecOffload.
>
> Equation 1 presents the overall optimization framework: maximizing system throughput while ensuring storage usage does not exceed the defined limit. System storage and throughput are functions of (1) static device and model parameters (e.g., $T_{target,prefill}^{GPU}$, $V_{target,FFN}$) and (2) tunable inference parameters (e.g., $bs$, $n_{cand}$). The detailed derivations are in Appendix A.1. While device and model parameters are fixed by the hardware and model, inference parameters are adjustable. Therefore, solving Equation 1 involves finding the optimal inference parameters that maximize throughput.
>
> We acknowledge that the simplified form of Equation 4 may cause misunderstanding. The full, unsimplified expression is:
>
> $$T_{decoding}=\max(T_{target,FFN}^{C2G},T_{draft})+T^{GPU}_{target,FFN}$$
>
> In this equation, $T_{target,decoding}$ (the target model's decoding work) consists of I/O time $T_{target,FFN}^{C2G}$ and compute time $T_{target,FFN}^{GPU}$. As our pipeline (Fig. 4) illustrates, in an offloading context, the I/O time $T_{target,FFN}^{C2G}$ dominates the computation time $T_{target,FFN}^{GPU}$.
>
> Therefore, we approximated the decoding cost as $T_{target,decoding} \approx T_{target,FFN}^{C2G}$ (see Eq. 18 for analysis) and omitted the negligible $T_{target,FFN}^{GPU}$ term from the $\max()$ function for brevity.
>
> Since our design *parallelizes* the parameter I/O (represented by $T_{target,decoding}$) and the draft model computation ($T_{draft}$), the total latency is determined by the $\max()$ of these two concurrent operations, not their sum. This explains the simplified form presented in Eq. 4.
>
> # 5. Necessity and function of the planner
>
> > It is not clear what is the search space of the planner. It seems everything is deterministic, or can be defined very easy. So the planner seems not necessary.
>
> As stated in Q4, we formulate throughput maximization as an optimization problem. The objective is to maximize throughput, the constraint is GPU memory capacity, and the search space is SpecOffload's inference parameters. This complex optimization problem necessitates the Planner.
>
> Specifically, the search space includes: the prefill batch size ($bs_{prefill}$), the decoding batch size ($bs$), the draft model's inference batch size ($bs_{draft}$), and the number of draft tokens ($n_{cand}$).
>
> For instance, Appendix A.3.1 shows that multiple *feasible solutions* (parameter combinations) can satisfy the memory constraint. However, these feasible solutions yield different inference throughputs. The Planner's objective is to search this space to find the optimal feasible solution that maximizes throughput.

---

### Author Response · Authors · 2025-12-01
**Summary of Contributions and Rebuttal Updates**

We thank the reviewers for their constructive feedback and are encouraged by the consensus on our work's core value. Specifically, Reviewer **F4bn** found our insight regarding the "marginal utility of GPU memory" to be "inspiring" , and Reviewer **255g** credited us for targeting the "under-explored inefficiency" of GPU idleness during offloading. Regarding the methodology, Reviewer **D7ni** commended the "clever method" of leveraging these execution bubbles , while Reviewer **7n7D** recognized our approach as a "clear nice followup" for better GPU utilization. Furthermore, Reviewers **255g** and **F4bn** praised the "coherent system" design and the "clear and informative" presentation.

Given the mixed scores, we would like to briefly summarize our core contributions and how we addressed the primary concerns regarding novelty and usage scenarios.

1. Core Insight: Transforming the "Offloading Paradox" into Opportunity. A primary concern (Reviewer F4bn) was whether SpecOffload is merely an engineering integration. We respectfully argue that our contribution is a fundamental shift in how offloading is approached.
   * First, we identify a counter-intuitive paradox in existing offloading frameworks: despite being designed for GPU constraints, they suffer from low GPU core utilization due to I/O bottlenecks, while large GPU memory usage offers limited performance gains. This profound diagnosis of "latent resources" is a key insight that has been overlooked by prior work.
   * Second, we propose a new paradigm that embraces the I/O bottleneck rather than hiding it. By embedding speculative decoding into the offloading pipeline, we convert idle compute and underutilized memory during I/O waits into productive work for the draft model—unlocking latent GPU capacity with near-zero overhead.

2. Validity of the "Large-Batch on Edge" Scenario Reviewers F4bn and 7n7D questioned the realism of running large batches on a single GPU.
   * As detailed in our rebuttal, this setting is critical for "Back-of-House" offline tasks where throughput is the sole metric. Examples include local synthetic data generation, privacy-preserving document processing (e.g., financial invoices), and RAG database indexing.
   * This focus aligns with recent high-throughput works on constrained hardware (e.g., FlexGen[ICML'23], Klotski [ASPLOS'25], MoE-Lightning [ASPLOS'25]).

3. Robustness and Generalization (Rebuttal Updates). During the rebuttal, we provided extensive new data to address specific concerns:
   * Planner Generalization: We demonstrated that our ParaSpec Planner adapts to new hardware/models in <5 minutes with minimal overhead (<10s runtime) [Response to 255g, 7n7D].
   * CPU vs. GPU Drafting: We verified that even with AVX-512 enabled, CPU-based drafting (e.g., 18 tokens/s) is orders of magnitude slower than our GPU-based approach (423 tokens/s), confirming that offloading the draft model to the CPU creates unacceptable pipeline bubbles [Response to 7n7D].
   * Acceptance Rates: We provided sensitivity analyses showing SpecOffload maintains performance gains across varying draft token acceptance rates [Response to 7n7D].

Conclusion: SpecOffload achieves 4.49x higher GPU utilization and 2.54x throughput gain over state-of-the-art baselines. We believe this work offers a timely and practical solution for democratizing LLM inference on commodity hardware.

Thank you for your time and consideration.

---

### Meta-Review · Area_Chair_sg7r · 2026-01-11

**Summary:**

SpecOffload proposes embedding speculative decoding into the offloading pipeline to utilize idle GPU resources during CPU-GPU I/O. The system comprises three components: dual-batch interleaved execution, adaptive tensor placement, and a ParaSpec planner. While reviewers acknowledged the important problem setting and the "inspiring" insight about marginal utility of GPU memory, significant concerns remain about novelty and experimental design. The primary weakness is that the baselines lack speculative decoding, making it unclear whether gains come from the proposed integration or simply from adding speculative decoding itself. The ablation study ("Serial SD" vs "SpecOffload") partially addresses this, but the method appears to be primarily an engineering integration of existing techniques. The three-part system design lack clarity, and the usage scenario (large batch inference on single commodity GPU) was questioned as unrealistic by multiple reviewers.

**Reviewer Concerns:**

Partially addressed: (1) Pipeline details clarified with per-layer explanation; (2) Planner generalization demonstrated across 4 hardware setups in <10s runtime; (3) CPU vs GPU drafting compared (CPU too slow even with AVX-512); (4) Acceptance rate sensitivity analysis added. Outstanding: (1) The fundamental novelty concern—whether gains primarily come from adding speculative decoding rather than the integration—remains contentious; F4bn explicitly maintained their score citing incremental contribution; (2) The usage scenario (large batch on single GPU) was still considered unrealistic by F4bn despite authors citing precedent from FlexGen/Klotski/MoE-Lightning; (3) 7n7D raised valid concern about AMX instructions for CPU drafting that wasn't fully resolved.

**Reviewer Scores:**

D7ni (6): Did not respond post-rebuttal; pipeline clarifications likely satisfied concerns, score unchanged. 7n7D (4): Asked follow-up questions about CPU/AMX and dynamic scenarios; concerns not fully resolved, score likely unchanged. 2S5g (6): Did not respond post-rebuttal; novelty concern acknowledged but not central to their review, score likely unchanged. F4bn (4): Explicitly maintained score post-rebuttal, stating contribution is incremental and scenario unrealistic.

---

### Decision · Program_Chairs · 2026-01-26

Reject